# Adjacent single-atom irons boosting molecular oxygen activation on MnO$_2$

Huayu Gu[1], Xiao Liu[1✉], Xiufan Liu[1], Cancan Ling[1], Kai Wei[1], Guangming Zhan[1], Yanbing Guo[1] & Lizhi Zhang ![ORCID] [1✉]

Efficient molecular oxygen activation is crucial for catalytic oxidation reaction, but highly depends on the construction of active sites. In this study, we demonstrate that dual adjacent Fe atoms anchored on MnO$_2$ can assemble into a diatomic site, also called as MnO$_2$-hosted Fe dimer, which activates molecular oxygen to form an active intermediate species Fe(O = O)Fe for highly efficient CO oxidation. These adjacent single-atom Fe sites exhibit a stronger O$_2$ activation performance than the conventional surface oxygen vacancy activation sites. This work sheds light on molecular oxygen activation mechanisms of transition metal oxides and provides an efficient pathway to activate molecular oxygen by constructing new active sites through single atom technology.

[1] Key Laboratory of Pesticide & Chemical Biology of Ministry of Education, College of Chemistry, Central China Normal University, 430079 Wuhan, P. R. China. ✉email: liuxiao71@mail.ccnu.edu.cn; zhanglz@mail.ccnu.edu.cn

Molecular oxygen activation, a continuous process of adsorption and dissociation of $O_2$ on the catalyst surface (Supplementary Fig. 1), is a key step in catalytic reactions[1], including the synthesis of organic compounds, catalytic combustion of volatile organic compounds (VOCs), oxygen reduction reaction (ORR) in fuel cells and so on[2–4]. The construction of oxygen vacancy, as an anion defect, is widely studied, which could improve the ability of molecular oxygen activation by electron transfer from the surface to the adsorbent[5,6]. However, high concentration of oxygen in air or high temperature will inevitably lead to the refilling of oxygen vacancy[7]. Although the production of surface unsaturated oxygen atoms with dangling bonds by surface chemical modification is a pathway to provide reactive oxygen species, the limited surface oxygen contents hinder the continuous reaction due to the inadequate supply of reactive oxygen species[8]. The use of metalloenzymes to generate different metal-oxo species can also trigger many oxidation reactions[9], but suffers from their high cost. Therefore, facile and efficient dioxygen activation is still a bottleneck for catalytic oxidation reaction[4].

The application of single atom technology in molecular oxygen activation paves a new pathway[10]. For example, Wu et al. reported that negatively charged Au atoms could activate molecular oxygen by regulating the depletion of adjacent oxygen atoms to produce defect oxygen sites in single-atom catalyst Au/CuO[11]. Tang and coworkers proposed that Na atoms could provide electrons to the catalyst surface, and thus endowed lattice oxygen with higher electron density to more easily participate in the catalytic reaction[12]. These examples revealed that single atoms promoted molecular oxygen activation via activating surface or lattice oxygen of metal oxides. However, it is still unknown whether the single atoms on transition metal oxides are directly involved in molecular oxygen activation process. Meanwhile, the difference in activation mechanisms between single atom/transition metal oxide and conventional surface oxygen vacancies based on transition metal oxide catalysts is also unclear[13].

To tackle these two issues, hereby, we anchored single Fe atoms on $MnO_2$ (Fe/$MnO_2$) through an oxalate-chelation-assisted hydrothermal method. Compared with conventional oxygen vacancies in $MnO_2$, two adjacent single-atom Fe sites of Fe/$MnO_2$ are more benefit to activate oxygen molecules immediately by forming Fe(O＝O)Fe active species. From a molecular level, pendant oxygen atom from Fe(O＝O)Fe active species with end-on mode was endowed weak bond strength and charge localization to participate in the reaction with lower activation barrier than $O_2$ activated in oxygen vacancy of pure $MnO_2$. The elucidation of two adjacent single-atom active sites enables further opportunities for the inert molecular activation.

## Results

**Chemical structure characterization**. We selected Fe/$MnO_2$ with 0.25% theoretical Fe content for systematic characterization. All the characteristic X-ray powder diffraction (XRD) peaks of Fe/$MnO_2$ well matched the standard card (JCPDS 44-0141) without additional impurity peaks (Supplementary Fig. 2)[14]. The high-angle annular dark-field scanning transmission electron microscope (HAADF-STEM) images of Fe/$MnO_2$ revealed (010) plane was the main exposed surface (Fig. 1a), which was used in the following calculation and simulation, and no cluster appeared on the catalyst surface (Fig. 1b). The energy-dispersive X-ray spectroscopy (EDXS) mapping suggested even distribution of Fe, Mn and O elements on the catalyst surface (Fig. 1c). X-ray absorption near edge structure (XANES) and extended X-ray absorption fine structure (EXAFS) spectra of Fe (Fig. 1d, e and Supplementary Fig. 3) displayed that Fe was uniformly distributed on the surface

of $MnO_2$ but not in clusters. Supplementary Fig. 4 was XANES and EXAFS spectra of Mn. Regarding to the Mn–O characteristic peak at 1.52 Å, Fe/$MnO_2$ with lower Mn–O intensity was ascribed to either the substitution of Mn or the formation of more oxygen defects. Compared with $MnO_2$, the Mn–O bond length in Fe/$MnO_2$ did change, indicating that Fe may not enter the interior of the bulk to change the coordination environment of Mn (also observed in Supplementary Fig. 4b). Referring to the Fe K-edge EXAFS spectra, the length of the Fe–(O)–Mn bond in the second shell was consistent with the appearance of the Mn–(O)–Mn bond, illustrating that Fe was located at the site of Mn, but had negligible effect on Mn. Similar results were obtained in the study of Smith et al.[15]. According to the results of wavelet transform (Supplementary Fig. 5), the coordination environment of Fe element in Fe/$MnO_2$ was obviously different from that of reference sample $Fe_2O_3$, especially in the second coordination layer. We also compared the Fe 3$d$ XPS spectra of $Fe_2O_3$ and Fe/$MnO_2$ (Supplementary Fig. 6), and found that the characteristic peak of Fe/$MnO_2$ had lower binding energy (710.6 eV) than that (711.4 eV) of $Fe_2O_3$. Meanwhile, Bader charge calculation results revealed that Fe on Fe/$MnO_2$ possessed more electrons than that of $Fe_2O_3$, consistent with the results of XANES absorption (Fig. 1d). Therefore, the distribution of Fe in the form of $Fe_2O_3$ on the catalyst surface can be excluded.

**Catalytic activity test**. High-efficiency CO oxidation is of great significance in automotive exhaust purification and the antitoxicity improvement of proton exchange membrane fuel cells[16,17]. Thereby, CO oxidation was taken as a probe reaction to check molecular oxygen activation performance of Fe/$MnO_2$ and $MnO_2$. Among the six Fe atoms anchored $MnO_2$ samples, Fe/$MnO_2$ with 0.25% theoretical Fe content achieved the highest catalytic activity under the conditions of 1% CO, 4% $O_2$ and argon as equilibrium gas (Supplementary Fig. 7a). Fe/$MnO_2$ could realize 100% conversion of CO at 80 °C, much better than $MnO_2$ of 100% CO conversion at 120 °C (Fig. 2a). Meanwhile, we calculated the specific activity ($R_{T50}$) of 0.25% Fe/$MnO_2$ according to the calculation formula (10) and the value was 310 $mol_{CO}$ h$^{-1}$ $mol_{Fe}$$^{-1}$ with $T_{50}$ at 47 °C, higher than those of most platinum group metal (PGM) catalysts[18]. Impressively, Fe/$MnO_2$ maintained stable catalytic activity at 80 °C for 50 h (Supplementary Fig. 7b), and robustly resisted the transformation between humid and dry atmospheres (Supplementary Fig. 7c). In contrast, $MnO_2$ seriously suffered from water vapor poisoning with a gradually decreasing conversion rate. Meanwhile, higher surface area normalized catalytic activity of Fe/$MnO_2$ ruled out the contribution of surface areas to its enhanced activity (Supplementary Figs. 8 and 9)[19].

To further explore the catalytic process, the Arrhenius plots of Fe/$MnO_2$ and $MnO_2$ were obtained (inset in Fig. 2a). The apparent activation energy ($E_a$, 33.7 kJ mol$^{-1}$) of Fe/$MnO_2$ was much lower than that (70.1 kJ mol$^{-1}$) of $MnO_2$, indicating a lower CO oxidation activation barrier of Fe/$MnO_2$ and different CO oxidation mechanisms of Fe/$MnO_2$ and $MnO_2$. Subsequently, we investigated the effects of partial pressures of CO and $O_2$ on turnover frequence (TOF) in the low-temperature region (Fig. 2b and c; Supplementary Fig. 10). The CO and $O_2$ reaction orders (0.01 and 0.35) of Fe/$MnO_2$ were significantly lower than those (0.09 and 0.61) of $MnO_2$. The higher reaction order of $O_2$ than CO illustrated that the $O_2$ activation was a key step, consistent with the previously reported results[20,21]. Obviously, Fe/$MnO_2$-activated molecular oxygen more easily than $MnO_2$, accounting for its better CO oxidation catalytic activity.

**Detection of active species**. We thus distinguished the oxygen species generated during molecular oxygen activation by $O_2$-TPD

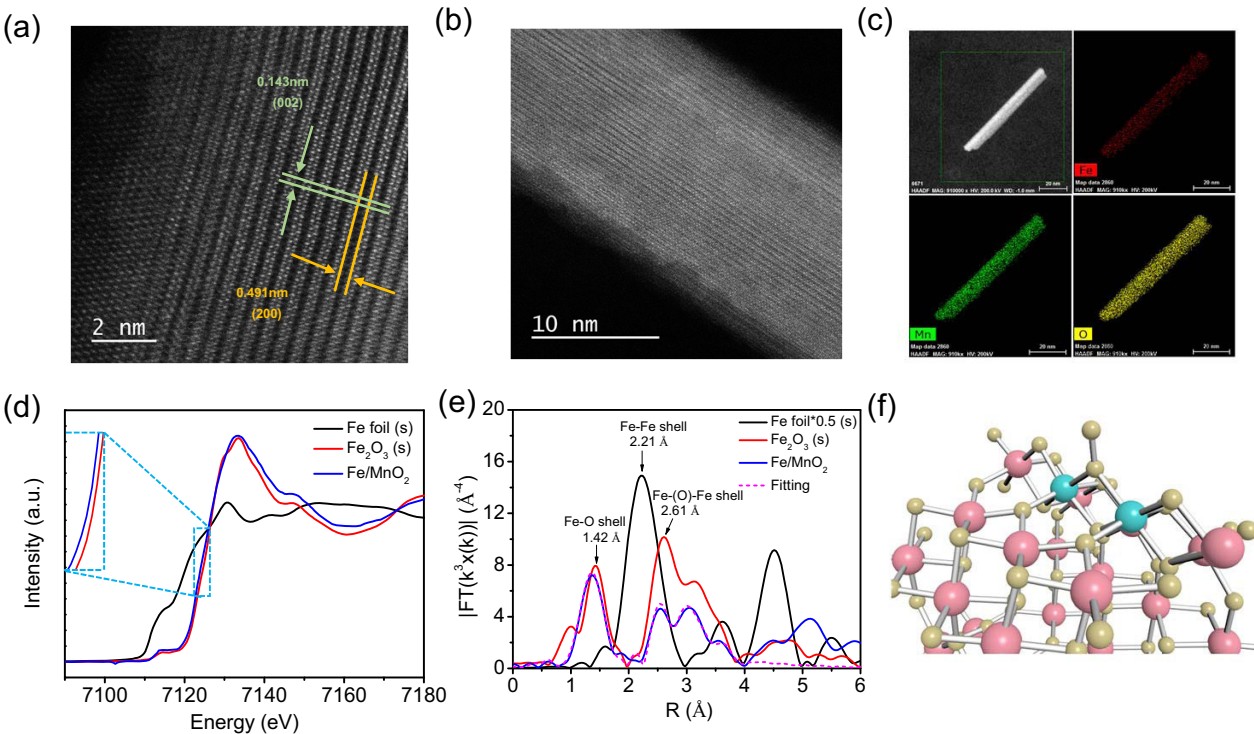

**Fig. 1 Structure characterization of Fe/MnO₂. a** and **b** HAADF-STEM images of Fe/MnO₂ at different magnifications. **c** Element mapping images of Fe/MnO₂. **d** Normalized XANES spectra of Fe. **e** Fourier-transformed K-edge EXAFS spectra in R-space of Fe (without phase correction) and the fitting of Fe/MnO₂ structure in (**f**).

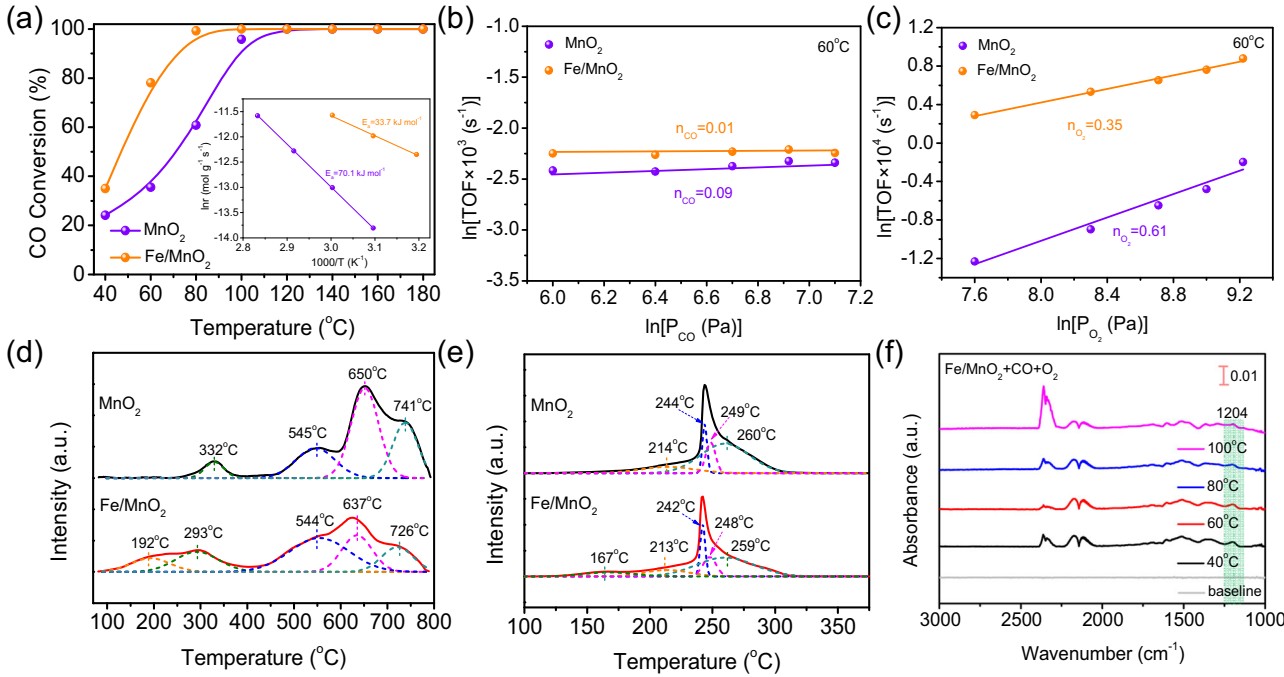

**Fig. 2 Detection of active species. a** Light-off curves of CO oxidation over Fe/MnO₂ and MnO₂ (the inset shows associated Arrhenius plots). Effect of (**b**) CO and (**c**) O₂ partial pressure on TOF of supported Fe/MnO₂ and MnO₂ at 60 °C, respectively. **d** O₂-TPD profiles and **e** H₂-TPR profiles of Fe/MnO₂ and MnO₂. **f** In-situ DRIFTS of CO oxidation over Fe/MnO₂ in a continuous flow of 1% CO/4% O₂/N₂ at different temperatures.

(Fig. 2d). The O₂-TPD profile of MnO₂ could be deconvoluted into four peaks of <400, 400–550, 550–700 °C and above 700 °C, corresponding to surface adsorbed oxygen ($O_{ad}$), surface lattice oxygen (S-$O_{latt}$) bound to $Mn^{3+}$, S-$O_{latt}$ and bulk lattice oxygen (B-$O_{latt}$) bound to $Mn^{4+}$, respectively[22]. Besides these four peaks,

an additional characteristic peak at 192 °C appeared in the spectrum of Fe/MnO₂, suggesting the emergence of new oxygen species formed on the surface. The formation of new oxygen species in Fe/MnO₂ was further confirmed by the new characteristic peak located at 167 °C in H₂-TPR (Fig. 2e) and the

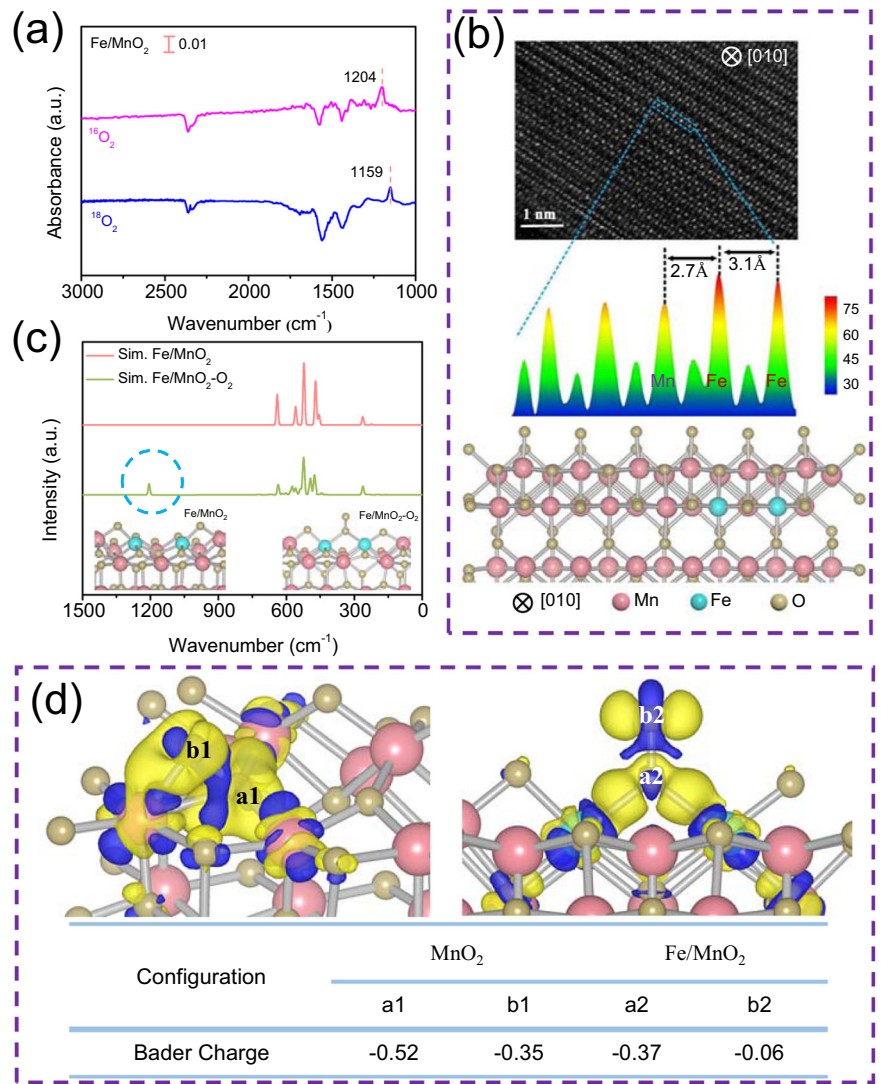

**Fig. 3 Detection of active sites. a** In-situ DRIFTS of $^{18}O_2$ isotope over Fe/MnO$_2$ in the out-line system. **b** STEM image of Fe/MnO$_2$ (top), intensity surface plot from blue dashed rectangle (middle) and the corresponding structural model (bottom). **c** simulated infrared spectra of Fe/MnO$_2$ and Fe/MnO$_2$ with O$_2$ adsorption. The bottom of the picture was the structure of the different species. **d** The difference charge density of O$_2$ adsorbed on MnO$_2$ (on the upper left) and Fe/MnO$_2$ (on the upper right). Different oxygen atoms from adsorbed oxygen species were labeled as **a** and **b**, and corresponding Bader charge was recorded in the bottom of charge density map. The charge density of yellow and blue represents the concentrated and scarce electrostatic potential scale respectively.

increased adsorbed oxygen species on Fe/MnO$_2$ surface according to the result of O 1$s$ XPS (Supplementary Fig. 11, Supplementary Table 1), respectively. These experimental results indicated that the excellent catalytic activity of Fe/MnO$_2$ might be attributed to the generation of new active oxygen species.

Subsequently, in-situ diffuse reflectance infrared Fourier transform spectroscopy (DRIFTS) was employed to confirm the generation of new oxygen species during CO oxidation over Fe/MnO$_2$. After only passing CO through the surface of the catalyst, gas phase CO peaks appeared at 2172 and 2113 cm$^{-1}$ (Supplementary Fig. 12). The peaks at 1520 and 1365 cm$^{-1}$ were ascribed to the symmetric and antisymmetric expansion vibration of carbonate, and others at 2361 and 2335 cm$^{-1}$ were originated from the vibration of CO$_2$ produced on the catalyst surface. Interestingly, a new characteristic peak arose at 1204 cm$^{-1}$ in case of CO and O$_2$ co-passing (Fig. 2f). When $^{16}O_2$ was replaced by $^{18}O_2$, this characteristic peak shifted from 1204 cm$^{-1}$ to the lower wavenumber of 1159 cm$^{-1}$ due to the isotopic effect (Fig. 3a),

indicating that this peak was closely related to the new active oxygen species generated via O$_2$ activation on Fe/MnO$_2$[23].

To determine the configuration of oxygen species, we compared the adsorption energy of different possible configurations of O$_2$ adsorbed on Fe/MnO$_2$ (Supplementary Fig. 13a–e), and interestingly found that Fe(O = O)Fe formed via bridging two adjacent Fe atoms with O$_2$ in end-on mode possessed the highest adsorption energy, indicating that this new structure of Fe(O = O)Fe was the preferentially adsorbed configuration of O$_2$. The existence of adjacent Fe atoms was thus validated by the STEM image of the (010) exposed Fe/MnO$_2$ catalyst (the top of Fig. 3b). According to the STEM test on bare MnO$_2$ (Supplementary Fig. 14), the intensity and the distance between the atoms (2.9 Å) are in accordance with the theoretical calculation model of MnO$_2$[24]. As shown in the corresponding selection-area intensity surface plot (the middle of Fig. 3b), the different peak intensities corresponded to the atomic number, so the higher and lower peaks are attributed to Fe and Mn[25],

respectively. Therefore, the distance between two adjacent Fe atoms is 3.1 Å, larger than that (2.7 Å) between Mn and Fe, consistent with the results of DFT calculation results (the bottom of Fig. 3b). Based on the analysis of element intensity in different positions of the STEM image, many two adjacent single-atom Fe sites were distributed on the surface of $Fe/MnO_2$ (Supplementary Fig. 15). Subsequently, we statistically analyzed the STEM spectra of 300 atoms in three regions on the surface of different $Fe/MnO_2$ catalysts (Supplementary Figs. 16, 17 and Supplementary Table 2), and found that 80.9% of Fe (the number of adjacent Fe atoms ≥ 2) on the surface were distributed as adjacent Fe sites for 0.25% $Fe/MnO_2$, with a small number (19.1%) as monatomic Fe site. Obviously, these adjacent Fe sites strongly contributed to the efficient activation of molecular oxygen. Meanwhile, the fitting result (dot line in Fig. 1e) of the model established by DFT simulation (Fig. 1f) well overlapped with the experimental measurement results within the range of 4 Å from scattering atoms in the EXAFS spectrum of Fe. Furthermore, we simulated the infrared spectra of $Fe/MnO_2$ and $Fe/MnO_2$ with $O_2$ adsorption by DFT calculation (Fig. 3c). In the simulated infrared spectrum of $Fe/MnO_2$ with $O_2$ adsorption in end-on mode, a new characteristic peak appeared near 1200 $cm^{-1}$, which was consistent with the experiment results. We also calculated the situation for two oxygen atoms of oxygen molecule attached to two Fe sites in side-on mode (Supplementary Fig. 13b). Two oxygen atoms got the same number of electrons, and the electron density was evenly distributed among the adsorbed oxygen molecules on the basis of the results of Bader charge and the differential charge density (Supplementary Fig. 18). Therefore, the dipole moment of oxygen molecules did not change during the vibration process in case of side-on mode, inconsistent with the results of the in-situ DRIFTS, thus further confirming the formation of Fe(O=O)Fe species[26].

The role of Fe(O=O)Fe on the $O_2$ activation was further checked by in-situ DRIFTS. The peak strength of Fe(O=O)Fe species at 1204 $cm^{-1}$ increased along with passing $O_2$ over $Fe/MnO_2$ (Supplementary Fig. 19a). When $O_2$ was changed to CO, this peak became weakened and finally disappeared, with generating $CO_2$ and carbonate species on the catalyst surface (Supplementary Fig. 19b). Therefore, we proposed that Fe atoms served as the active site for molecular oxygen activation, producing Fe(O=O)Fe to oxidize CO. In order to further confirm the mode of Fe(O=O)Fe participation in the CO oxidation, $Cl_2$ was introduced into the oxidation system[27–29]. For $Fe/MnO_2$ with pre-injecting $O_2$ and $Cl_2$ successively, the amount of Fe(O=O)Fe and intermediate species did not change even after 30 min when CO was added to the reactor (Supplementary Figs. 20, 21), indicating that $Cl_2$ occupied CO adsorption sites and prevented Fe(O=O)Fe from participating in the reaction. Therefore, Fe(O=O)Fe species might react with adsorbed CO through Langmuir–Hinshelwood (L–H) mechanism, different from surface oxygen vacancies ($V_O$) of $MnO_2$ via the Mars–van Krevelen (MvK) reaction mechanism[30].

**Oxygen activation mechanism**. To better understand the $O_2$ activation and CO oxidation performance of Fe(O=O)Fe, we compared the $O_2$ adsorption in $Fe/MnO_2$ with two adjacent single-atom Fe sites and $MnO_2$ with $V_O$. The adsorption energy (−2.00 eV) of bridging two adjacent Fe atoms with $O_2$ was relatively higher than that (−0.83 eV) of $O_2$ adsorption over $V_O$ (Supplementary Fig. 13). Subsequently, the bonding situation of $O_2$ adsorbed in two different sites was further compared by differential charge density maps. As shown in Fig. 3d, the oxygen atoms a1 and b1 (from the adsorbed oxygen molecules on the $V_O$) are bound with the unsaturated Mn sites in $MnO_2$. The

strong electron scale is distributed among the valence bonds, indicating the formation of a firm chemical bond between oxygen atoms and Mn sites. The feedback π electrons of Mn d orbital are transferred to the vacant orbital of oxygen, resulting in the higher charge density of oxygen atoms. The correlation results are proved by Bader charge (bottom of Fig. 3d). For $Fe/MnO_2$, oxygen atom a2 gets the electrons from two adjacent double Fe atoms with the Bader charge of −0.37, while oxygen atom b2 is weakly bonded with a2 becasue of scarce electrostatic potential scale located at O–O bond. We also calculated crystal orbital Hamilton populations (COHP) of $O_2$ adsorbed on $MnO_2$ and $Fe/MnO_2$ (Fig. 4a–c). Integral crystal orbital Hamilton populations (ICOHP) results (Fig. 4d) showed that the intensity of O–O bond from $O_2$ adsorbed on the double Fe sites was weaker than that on $MnO_2$ (–ICOHP = 2.73/2.44 vs. –ICOHP = 3.50/3.18 for $MnO_2$). So, b2 with the end-on mode of $O_2$ on adjacent Fe sites might easily escape to participate in the subsequent reaction, accounting for the facile molecular oxygen activation of two adjacent single-atom Fe sites with the lower activity barrier.

Furthermore, we compared the activation behavior of oxygen at Fe–Fe or Fe–Mn bimetallic sites, and found that the double Fe sites mainly contributed to the molecular oxygen activation of $Fe/MnO_2$ (Supplementary Fig. 22). Importantly, the oxalate-chelating coordinated hydrothermal method enables the uniform dispersion of monatomic Fe on the surface of the catalyst, constructing abundant active sites composed of adjacent Fe atoms (Supplementary Figs. 23–26). The results are also verified by etching XPS spectra (Supplementary Fig. 27).

DFT calculation was thus carried out to calrify the contribution of two adjacent single-atom Fe sites to the CO oxidation on $Fe/MnO_2$ (Fig. 5). First, CO was adsorbed on the Mn site[26], which was connected to Fe(O=O)Fe species with lattice oxygen. Being of the electronegativity, oxygen atom (b2) in Fe(O=O)Fe was inclined to bond with carbon atom in CO (Supplementary Fig. 28). After overcoming the activation barrier of 0.17 eV, $CO_2$ was formed and desorbed, leaving behind another O atom (a2). Subsequently, the lattice oxygen connected to the Mn site reacted with the second adsorbed CO to produce another $CO_2$ molecule by overcoming the activation energy of 0.51 eV. Then the remained oxygen atom (a2) between two adjacent Fe sites moved to refill the position of $V_O$. Finally, the surface structure of $Fe/MnO_2$ returned to its original state through the re-adsorption of $O_2$, realizing a full catalytic cycle of CO oxidation. Furthermore, we also calculated the energy of CO reacting with $O_2$ on different sites. The calculation results of the transition state revealed that the energy (0.19 eV) for CO reacting with $O_2$ adsorbed on the adjacent Fe sites was lower than that (0.37 eV) of $O_2$ adsorbed on the $V_O$ referring to Supplementary Fig. 29. Therefore, the MvK mechanism on bare $MnO_2$ could be inhibited by the high energy demand. As expected, the total CO oxidation reaction barrier (0.51 eV) of $Fe/MnO_2$ was much lower than that (0.86 eV) of $MnO_2$ (Supplementary Fig. 30), where $V_O$ governed both molecular oxygen activation and CO oxidation[31–34]. Two adjacent single-atom Fe sites on $MnO_2$ possessed significantly higher activity by forming Fe(O=O)Fe species than conventional $V_O$ to activate oxygen molecules.

## Discussion

In general, we constructed adjacent single-atom Fe-active sites to activate oxygen molecular by forming intermediate species Fe(O=O)Fe over $Fe/MnO_2$. The experimental and theoretical results revealed that $Fe/MnO_2$ exhibited lower reaction orders and lower activation barrier for oxygen activation than pure $MnO_2$ with oxygen vacancy as the activation sites. Charge

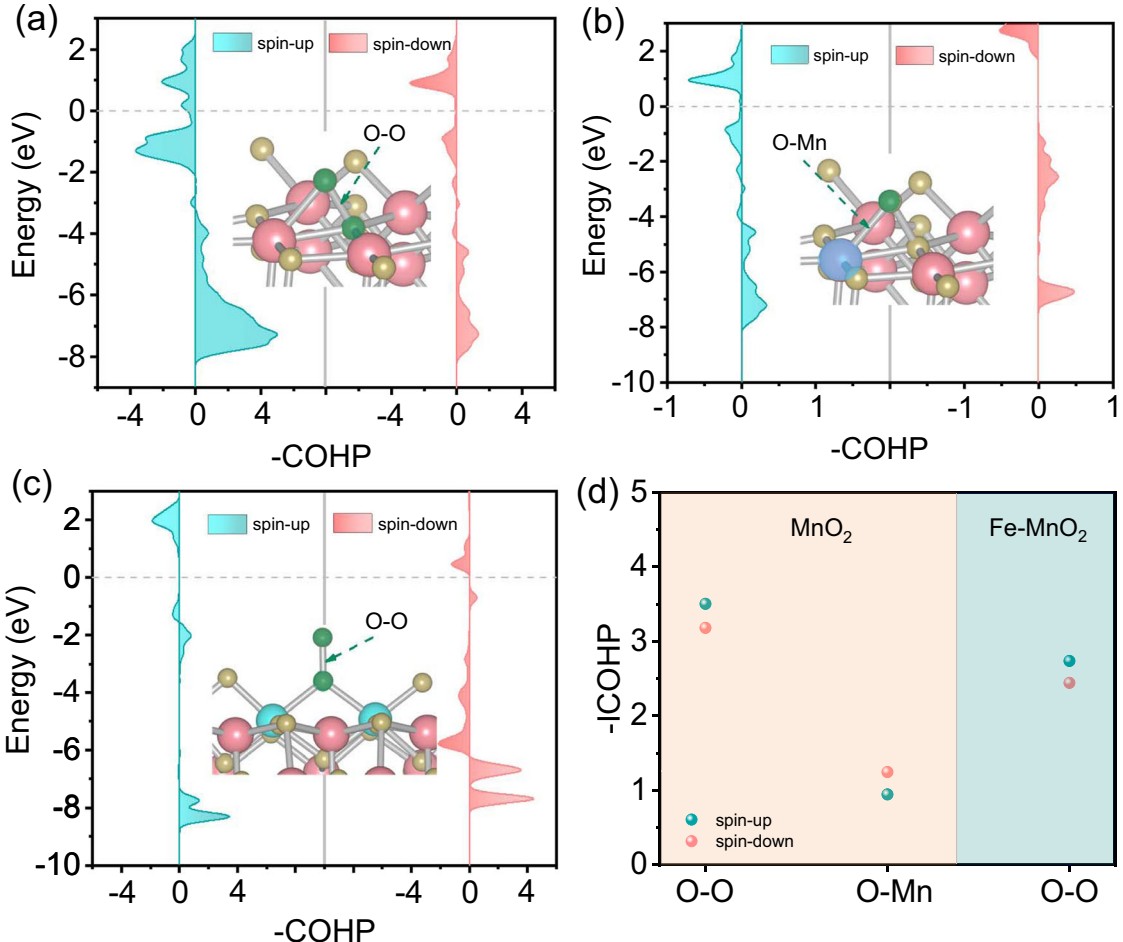

**Fig. 4 Analysis of bonding strength.** COHP of $MnO_2$ (**a**, **b**) and $Fe/MnO_2$ (**c**) at different positions. **d** The corresponding ICOHP in **a–c**. The insets in **a–c** show the corresponding structure.

localization and weak bond strength made pendant oxygen atoms more easily escape to participate in the subsequent oxidation reaction. This work offers dimer active sites for the dioxygen activation and opens up a path to promote oxidation reactions. More importantly, we believe that two adjacent single-atom active sites might be also more diverse and efficient for the activation of inert small molecules such as $N_2$, $CO_2$, $CH_4$, etc.

## Methods

All the reagents used in the synthesis process are analytical-grade and do not require further purification before use.

**Synthesis of Fe/MnO₂.** Fe/MnO₂ catalysts were prepared by conventional hydrothermal methods, a modified method previously reported elsewhere[35]. During the synthesis process, 1.58 g potassium permanganate (KMnO₄) solid was dissolved in 30 mL deionized water, and then 20 mL of 35.5 g L⁻¹ ammonium oxalate ((NH₄)₂C₂O₄·H₂O) solution was added to the potassium permanganate solution drop by drop. After 0.5 h of stirring, the ferric nitrate (Fe (NO₃)₃·9H₂O) solution was transferred to the above mixture solution and stirred again at room temperature for another 1 h. The mixture was transferred to Teflon bottle into a stainless-steel autoclave reacted for 24 h at 180 °C. After cooling to room temperature, the powder products were washed and filtered repeatedly, and dried at 105 °C for 12 h. The resulting gray-black powder was placed in a corundum crucible and calcined for 2 h at 350 °C under ambient air. Fe/MnO₂ catalysts with different contents were obtained by changing the mass of iron nitrate. The obtained catalyst was labeled as N% Fe/MnO₂, and N represented the ratio of Fe to MnO₂. The preparation of pure MnO₂ remained unchanged except that no iron nitrate was added to the precursor solution.

**Synthesis of Fe/MnO₂−100 °C.** The preparation process is the same as that of Fe/MnO₂, except that the hydrothermal reaction temperature is 100 °C instead of 180 °C and is not calcined at 350 °C.

**Synthesis of Fe₂(C₂O₄)₃−100 °C.** The ferric nitrate particle was dissolved in 30 mL deionized water and stirred for 30 min. The solution was added to 40 mL ammonium oxalate solution (35.5 g L⁻¹) one by one. After stirring for 1 h, the mixture was transferred to the oven and reacted at 100 °C for 24 h. After cooling to room temperature, the powder products were washed and filtered repeatedly and dried at 105 °C for 12 h.

**Catalysts characterization.** The as-prepared catalyst of X-ray diffraction (XRD) patterns is obtained through the Rigaku D/MAXRB diffractometer equipped with monochromatized Cu K alpha radiation, and scanning range of 2 theta angle changes from 20° to 80°. Aberration-corrected HAADF-STEM measurements are completed by using Titan transmission electron microscope operated at 300 keV. Different elements of energy-dispersive spectroscopy (EDX) were obtained on the same equipment. The actual element content of the catalyst was analyzed by an inductively coupled plasma optical emission spectrometer (ICP-OES) based on dissolved samples in aqua regia. The valences of different elements on the surface of the catalyst were analyzed by XPS (Thermo Scientific Escalab 250 Xi analyzer). The binding energy of each element was corrected based on the C1s spectrum at 284.8 eV. The XAFS test was conducted at the Beijing synchrotron radiation center. The sample was ground evenly and smeared into Scotch tapes. K-edge absorption of Fe and Mn elements was tested in fluorescent mode at room temperature. The samples were tested several times and the average results were further analyzed. Data is processed through[36]. The EXAFS spectrum of Fe was fitted in the R space by a theoretical model based on VASP optimization results according to the shorter absorption layer of Fe (including first Fe–O, Fe–(O)–Mn, and Fe–(O)–Fe shell). The corresponding fitting parameters are listed in Supplementary Table 3. The oxidation and reduction properties of samples can be compared through the hydrogen temperature-programmed reduction (H₂-TPR) and oxygen temperature-programmed desorption (O₂-TPD) experiments monitored by the Chemisorb tp-

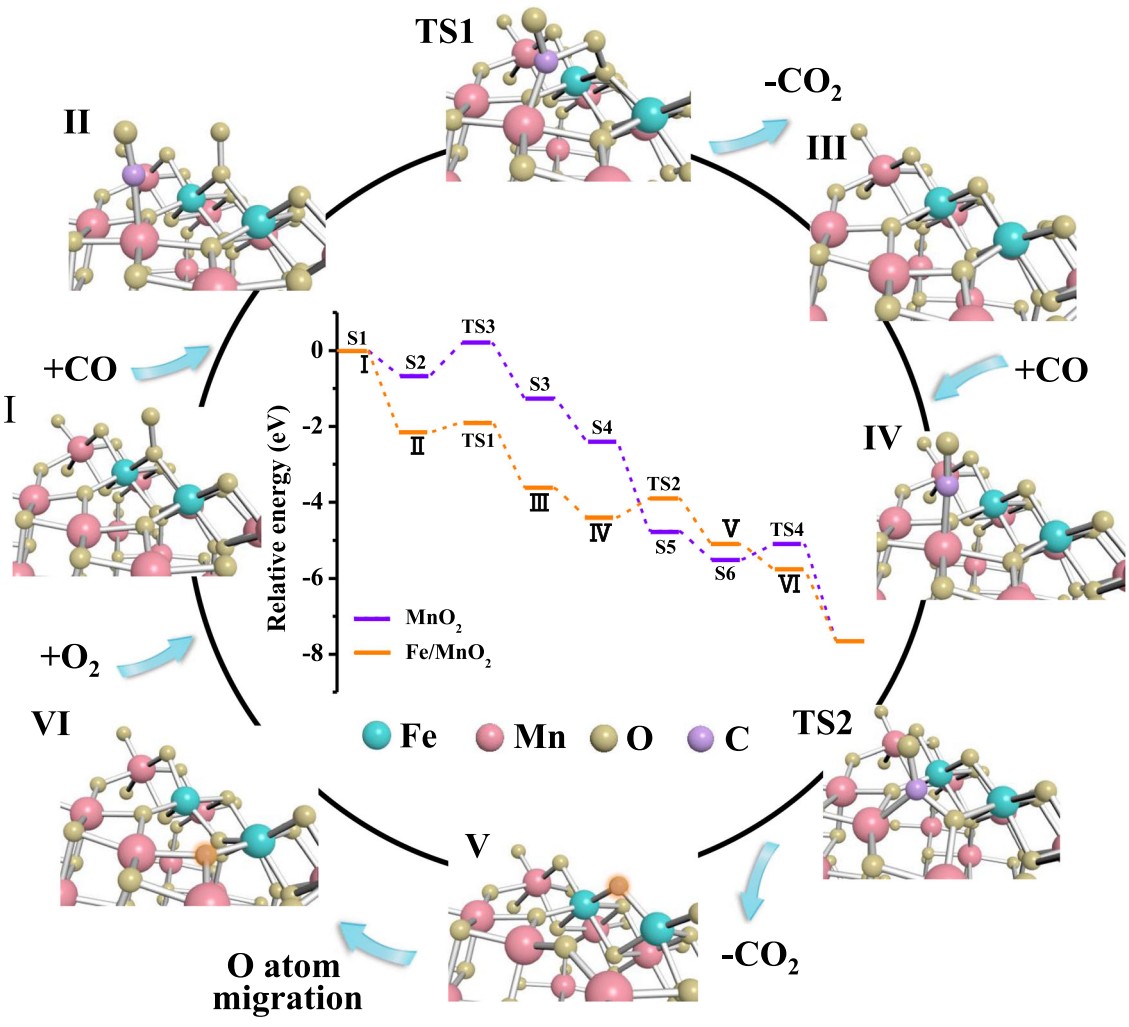

**Fig. 5 Mechanistic investigations.** Proposed reaction mechanism for CO oxidation on Fe/MnO$_2$ surface. The inset shows the calculated energy profiles in eV. The structures of intermediates and transition states are shown in the reaction cycle. The energy profile of the reaction cycle for Fe/MnO$_2$ is shown by the orange line in the inset. The purple line shows the energy profile for MnO$_2$ (reaction cycle is shown in Supplementary Fig. 30).

5080 with a thermal conductivity detector (TCD). Before the test, 100 mg catalyst was pretreated at 105 °C in an argon atmosphere for 30 min. After natural cooling to room temperature, measurement was started. For O$_2$-TPD, the samples were exposed to an atmosphere of 20% O$_2$/Ar (50 mL min$^{-1}$) for 1 h and then heated to 800 °C under N$_2$ atmosphere with 5 °C min$^{-1}$. For H$_2$-TPR, 5% H$_2$/Ar was continuously fed into the reactor containing the samples, and the temperature rose to 800 °C at a rate of 5 °C min$^{-1}$. The whole flow is recorded by TCD. During the catalytic reaction of CO oxidation, the in-situ reaction process of the catalyst surface was recorded in the DRIFTS (Nicolet iS50, Thermo). For the off-line system, the catalyst was treated at 300 °C for 30 min before testing to remove the oxidized species adsorbed on the catalyst surface. Then cooled naturally to room temperature, background noise was removed and the baseline was recorded. When the temperature was raised to the preset value, a certain volume of CO or O$_2$ was injected into the reaction device, and the spectrogram was collected. When the spectrogram no changed over time, it was the test result at this temperature. On the other hand, for the on-line system, the catalyst was pretreated at 300 °C for 30 min in the atmosphere of N$_2$, cooled to 40 °C, and deducted the background followed by collecting baseline. Accompanied by the temperature maintained at 40 °C, 1% CO/ N$_2$ and 1% CO/4% O$_2$/N$_2$ were continued to flow through the catalyst and the total flow rate was 50 mL min$^{-1}$, respectively. The in-situ spectrogram was recorded with time. For in-situ DRIFTS reaction involving Cl$_2$, after high-temperature pretreatment, 20 mL of Cl$_2$ was injected into a reactor containing MnO$_2$ at 40 °C. After the spectrum was stabilized, it could be observed that the peak of adsorbed species on the catalyst surface decreased significantly, and the characteristic peak of CO$_2$ was generated at the same time. Subsequently, 20 mL CO was injected into the closed system, and an obvious adsorption peak of CO appeared in the spectrum. The in-situ spectrogram was recorded for 30 min. As for Fe/MnO$_2$, 20 mL O$_2$ was injected into the vacuum reaction chamber at 100 °C, and the DRIFTS spectra were recorded immediately. After the temperature was lowered to 40 °C, 20 mL Cl$_2$ was injected into the reactor. The absorption peak of Cl$_2$ appeared in the spectrum.

Then, 20 mL of CO has added to the system again, and a significant vibration peak of CO appeared. When the spectrogram is stable, the corresponding results are recorded, respectively.

**Catalytic performance evaluation.** All catalyst was tested in a continuous flow reactor. The newly prepared catalyst of 100 mg was loaded into a quartz tube with an internal diameter of 1 mm, and 1% CO, 4% O$_2$, and 95% N$_2$ were fed into the reactor at the hourly space velocity (GHSV) of 30,000 mL g$^{-1}$ h$^{-1}$. The residual gas content composition was analyzed by online gas chromatography with a flame detector (FID). The temperature interval with 20 °C was used to study the relationship between different reaction temperatures and CO conversion performance. Test, at the same temperature many times, was recorded as the corresponding light-off conversion rates after the conversion effect remained stable. The conversion effect of CO is calculated using Eq. (1):

$$\eta\,(\%) = \frac{C_{in} - C_{out}}{C_{in}} \times 100\%　\qquad (1)$$

where $C_{in}$ and $C_{out}$ represent the CO concentration in the inflow and outflow reaction system, respectively, determined by gas chromatography.

At a constant temperature, the relationship between the exchange frequency (TOF) and the partial pressure of CO was tested. By keeping the partial pressure of O$_2$ unchanged at 4 kPa, the partial pressure of CO changed from 4 to 5.6 kPa. On the contrary, in order to explore the effect of partial pressure of O$_2$ on the reaction rate, the partial pressure of CO was kept unchanged at 4 kPa, and the partial pressure of O$_2$ was adjusted from 4 to 12 kPa.

The specific reaction rate ($R_{CO}$) was calculated by Eq. (2):

$$R_{CO} = \frac{v_{gas} \times \eta}{m_{cat}}　\qquad (2)$$

where $\upsilon_{gas}$ represents the molar gas flow rate (mol h$^{-1}$), and $m_{cat}$ is the mass of the catalyst (g).

Turnover frequency (TOF) was calculated by Eq. (3):

$$TOF = \frac{R_{gas} \times M}{\delta_{cat}} \qquad (3)$$

where $M$ represents the molar weight of Mn or Fe, and $\delta_{cat}$ represents the percentage of active sites from the catalyst. For $MnO_2$, the number of Mn sites (for Fe/$MnO_2$, the number of Mn and Fe sites) is calculated based on the ICP-OES results.

Arrhenius plots at different temperature ranges were tested by varying the catalyst mass (5–50 mg, diluted with quartz sand of 200 mg), CO concentration (2–2.4%), and carrier gas flow rates to ensure that CO conversion remained below 20%, thus eliminating the effects of heat and mass transfer on kinetic testing. The setting of relevant specific reaction conditions was listed in Supplementary Table 4.

The apparent activation energy ($E_a$) is calculated as follows:

$$k = Ae^{-E_a/RT} \qquad (4)$$

where $k$ is the rate constant, $A$ is the former factor, and $T$ is the temperature (K). $E_a$ is obtained by fitting the value of $\ln(r)$ at different temperatures.

Coverage of Fe atoms from Fe/$MnO_2$:

$$m_{Fe} = \frac{n_{Fe} \times M_{Fe}}{N_A} \qquad (5)$$

$$m_{Mn} = \frac{n_{Mn} \times M_{Mn}}{N_A} \qquad (6)$$

$$m_O = \frac{2 \times (n_{Mn} + n_{Fe}) \times M_O}{N_A} \qquad (7)$$

$$N = \frac{H}{h} \qquad (8)$$

$$W_{Fe} = \frac{2 \times m_{Fe}}{2 \times (m_{Fe} + m_{Mn} + m_O) + 300 \times m_{MnO_2} \times N} \qquad (9)$$

where $m$ represents the mass of different element; $N$ is the number of atoms; $M$ is the relative atomic mass; $N_A$ is the Avogadro constant; $H$ is the thickness of catalyst, which is counted by TEM test results. $h$ represents the height of single-layer $MnO_2$, which comes from the structural optimization model; $N$ is the number of layers; $W_{Fe}$ is the mass fraction of Fe.

Specific activity was calculated by Eq. (10):

$$R_{T50} = \frac{\upsilon_{CO} \times M_{Fe}}{m_{Fe}} \qquad (10)$$

where $\upsilon_{CO}$ represents the molar gas flow rate of CO (mol h$^{-1}$); $M_{Fe}$ is the relative atomic mass of Fe; $m_{Fe}$ is the mass of Fe on the basis of ICP results and statistical results from 0.25% Fe/$MnO_2$ STEM image.

**First-principles density functional theory (DFT) calculations**. All surface structures were simulated by the Vienna ab-initio simulation package (VASP), using GGA-PBE as the exchange-correlation functional, while the projector augmented wave (PAW) was utilized to handle the interaction of valence electrons with core and inner electrons[37–39]. The plane-wave basis restriction with 520 eV cutoff energy and the Hubbard $U$ for Mn$3d$ at 3.9 eV and Fe$3d$ at 3.0 eV are set for models[40]. Spin-polarization calculations were carried out for all systems. The convergence criteria for energy and force applied to all atoms are set at $10^{-5}$ eV and 0.02 eV Å$^{-1}$, respectively. The value of $k$ in the Brillouin zone is 3*3*1. Meanwhile, in order to avoid the interaction between layers caused by periodic boundary conditions, the thickness of the vacuum layer of the surface structure is set to 20 Å. According to the experimental results (STEM), the surface (010) of the 2*2 supercell $MnO_2$ or Fe/$MnO_2$ was used as the exposure of the catalyst to stimulate the reaction process. The transition state of the catalytic reaction was determined using the climbing image nudged elastic band (Cl-NEB) method[41,42]. Zero-point energy correction was obtained from vibrational frequencies by applying normal-mode analysis through DFT calculations[43]. We fixed the catalyst substrate and only allowed the adsorbing molecule to vibrate.

The adsorption energy was defined by

$$E_{ads} = E_{(ads+sur)} - E_{(ads)} - E_{(sur)} \qquad (11)$$

where $E_{(ads+sur)}$ is the total energy of a surface interacting with adsorbate, and $E_{(ads)}$ and $E_{(sur)}$ are the energies of the isolated adsorbate and clean surface, respectively. The energy is corrected by zero-point energy.

## Data availability
All data generated in this study are provided in the Article and Supplementary Information. The other data that support the findings of this study are available from the corresponding author upon reasonable request.

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

## Acknowledgements

We acknowledge the National Natural Science Foundation of China (21976066), National Key Research and Development Program of China (2019YFC1806203), the Fundamental Research Funds for the Central Universities (CCNU19QN065) for financial support.

## Author contributions

H.Y.G. and X.L. conceived and synthesized the catalyst and completed most of the experiments and characterization. X.F.L. completed the electron microscopy test. C.C.L. helped with the catalytic performance test. K.W. and Y.B.G. performed $H_2$-TPR and $O_2$-TPD tests. H.Y.G. and G.M.Z. carries out theoretical calculation and analysis. X.L. and L.Z.Z. designed and supervised the project. H.Y.G., X.L. and L.Z.Z. co-wrote the paper.

## Competing interests

The authors declare no competing interests.
