## [Peer Review File · Nature Communications]

Title: Adjacent single-atom irons boosting molecular oxygen activation on MnO₂REVIEWER COMMENTS

Reviewer #1 (Remarks to the Author):

In this manuscript, the authors have explored the CO oxidation process on the binuclear metal site of the heterogeneous Fe/MnO₂ catalyst. Sophisticated techniques and theoretic calculations have been employed to characterize the site, analyze its properties and explain the catalytic mechanism. It describes an interesting idea on how to activate oxygen with such a binuclear site in the complex catalysis process. However, the main conclusions are not convincing based on the provided experimental evidence. Thus I could not recommend its acceptance in its current form.

1. The concept of “Binuclear single-atom” in the title is really confusing and frustrating. It is common sense that each atom only has one nucleus. It seems that the term actually means a two-Fe-atom site or bi-atomic or “binuclear iron” site in the text. Why on earth do the authors use the “single-atom” concept?
2. The existence of binuclear iron sites is demonstrated with the HAADF-STEM images, but unfortunately other sites like single or triple iron atoms cannot be excluded. It's hard to believe that all iron atoms maintain the binuclear configuration. It seems that the manuscript selectively presents a few examples of neighboring binuclear iron atoms. Even so, one can still find that there is at least a single iron atom in Fig. S11 (4). Therefore the manuscript should provide the statistical distribution of the sites as a function of iron atom number inside.
3. It's not a good choice to directly compare the activities of Fe/MnO₂ and MnO₂ because typically a metal entity possesses a higher activity than an oxide support for the CO oxidation no matter how the metal atoms are dispersed on the oxide.
4. There is not strong evidence to prove the existence of the Fe(O=O)Fe intermediate. A new oxygen species is confirmed by O₂-TPD and H₂-TPR, but it could be any type of active oxygen species. The O1s XPS in Fig. S8c simply indicates the oxide peak and the wide feature of adsorbed oxygen is not any clue of surface-adsorbed O₂ species. Even if the oxygen adsorbs on the surface as O=O, there is no evidence to show that it sits in between two iron atoms.
5. The 1204 cm⁻¹ band in Fig. 3f is supposed to be relating to the Fe(O=O)Fe species. However, the authors should tell the readers that such a decrease in the wavenumber is indeed from the isotopic effect for the Fe(O=O)Fe species. The Fe(O=O)Fe species is non-polar, how could the O-O vibrational mode appear in IR? DFT could help calculate all vibrational modes of the proposed Fe(O=O)Fe species. Such calculations should be included.
6. An additional question about the blank experiment on the structure of the Fe/MnO₂ catalyst in Figure 1f: is there any oxygen molecule between the iron atoms at the binuclear site before O₂ adsorption takes place? According to the analysis in the manuscript, the active site should be Fe-(Ov)-Fe, then how can the O₂ molecule be adsorbed to form Fe(O=O)Fe? The EXAFS result does show the existence of Fe-O and Fe-(O)-Fe. Normally, Mn-O-Mn should be stable in the MnO₂ structure. Therefore, Fe-O-Fe should be more stable than Fe-(Ov)-Fe and Fe(O=O)Fe, then why is Fe-O-Fe non-existent in Fe/MnO₂?

Reviewer #2 (Remarks to the Author):

The authors claimed that they demonstrated that dual adjacent single Fe atoms anchored on MnO₂ can assemble into a binuclear site, which activates molecular oxygen to form an active intermediate species Fe(O=O)Fe for highly efficient CO oxidation. It was concluded that binuclear Fe sites exhibited a stronger O₂ activation performance than the conventional surface oxygen vacancy activation sites.

However, the data provided seems not enough to support their claims. I understood that Fe/MnO₂ showed better CO oxidation performance compared to pure MnO₂ but unfortunately not convinced the formation of binuclear Fe sites and L-H type reaction. The following is the reasons why I consider it is inadequate, including questions and comments. I hope this will help improve the quality of this author's work and contribute further understanding the physics behind this seminal catalytic reaction.

In Fig.1a, TEM with atomic resolution are shown and Fe and Mn atoms were identified by line profile in Fig. 3b. I assume that they were distinguished by the intensity differences of the bright spots from the line profile. If so, I suggest the authors to show TEM for MnO₂ to confirm that there is no intensity differences among the bright spots and prove intensity difference is inherently from Fe and Mn in Fe/MnO₂. I can see some bright spots on the surface of pure MnO₂ in HRTEM of Fig. 3a of the following literature, but their intensities are different.

J. Gao et al./Journal of Catalysis 341 (2016) 82-90.

I also suggest showing line profile for longer distance and calculating the Fe coverage to see if it is consistent to the amount of introduced Fe. In some areas, three or more bright spots appear to be lined up.

XAFS was taken for Fe foil (metallic Fe), Fe₂O₃ (Fe 3+) and Fe/MnO₂ and spectrum of Fe/MnO₂ is almost the same to that of Fe₂O₃. I suggest authors to discuss oxidation state of Fe based on the result of XPS of Fe on Fe/MnO₂ with literatures and Bader charge.

Fourier-transformed k-edge EXAFS for Fe₂O₃ and Fe/MnO₂ are quite similar. The possibility of formation Fe₂O₃ was excluded by comparing the Morlet wavelet transform in Supplementary Fig.3 but the results show no significant difference. The k³-weighted EXAFS can be judged to check the quality of the measured spectra but not shown. It may not be possible to prove that Fe₂O₃ is not formed only from the results of this XAFS analysis.

Peaks in Fig 3a is too small to see.

How can you exclude the possibility of MvK on bare MnO₂ surface of Fe/MnO₂? Is there any evidence that MvK is hindered by Fe atoms?

The Fe(O=O)Fe was suggested to the preferentially adsorbed configuration of O₂ since the adsorption energy is high, however, in the calculation in Supplementary Fig .10, Mn(O=O)Mn was not calculated.

What is the meaning of "molecular oxygen activation" and "activation behavior of oxygen" in line 32 and 181? Is it mentioning low dissociation energy of oxygen molecules? I suggest the authors define the

term and share the physical image clearly before it is used for the readership of this multidisciplinary journal.

In line 24, in introductory part, author described that high concentration of oxygen in air or high temperature will inevitably lead to the refilling of oxygen vacancy by referencing ORR study and not CO oxidation. I don't think this is appropriate citation. There are many studies for oxygen vacancy involved CO oxidation studies. I suggest authors to present discussions based on more relevant literatures. For the same reasons, reference 1 to 3 seems inappropriate (they are electrochemical ORR studies and there should be more relevant references).

Injection of Cl₂ was performed. Is it necessarily be Cl₂? Why Cl₂? I recommend authors to add explanation why this way is good for investigating/identifying the CO oxidation mechanism, probably with citing some articles.

In Supplementary Fig. 11 STEM images were shown but two of them are labeled the same. I recommend that you carefully check the other data to make sure there are no misrepresentations. Unit error was found in Line 108 and it should be 167 C, not cm⁻¹.

In Supplementary Fig. 6, fitted light off curve for MnO₂ is awkward. Fitting needs to be performed based on the provided data and how you fit needs to be mentioned with reliability factor.

Reviewer #3 (Remarks to the Author):

In this work, the authors report on a novel "single-atom" binuclear iron site embedded in a MnO₂ host lattice. Overall, this work is of very high quality and the findings are corroborated with theoretical study. This kind of earth-abundant material may well serve as the next generation of catalyst for a variety of chemical technologies and this type of fundamental study will be valuable in identifying paths forward in both improving and utilizing such structures. A few minor suggestions that might improve this manuscript:

- 1) While I understand the usage in comparison to previous work, the portion of the title "Binuclear single-atom irons" is a bit problematic as the binuclear and plural of iron both seem to contradict the "single-atom" term. This term is of course somewhat problematic for all "single-atom" sites since local structure can strongly influence key binding sites but even more so here where 2 distinct atoms are serving as the localized active site. A similar issue occurs at the end of the discussion on page 12 where the term "monatomic binuclear active sites" is used.
- 2) Oxygen activation is of course key to a wide variety of (electro)chemical reactions. A sentence on some of the most technologically important applications and some additional discussion on the importance of CO ⇌ CO₂ in particular would help highlight the importance of this work.
- 3) On a technical level, the denticity/binding motif of the bound O₂ as brought up on page 7 and Figure S10 could be strengthened somewhat. In particular, it seems odd to not have considered and reported

on the energetics of the O₂ with 1 O bonded to each of the Fe atoms, even if it is indeed higher energy than the O₂ bridging the 2 Fe with a single O. Also, based on equation 5, it appears that binding does not include vibrational contributions to the free energy including zero-point energies. Given how close some of the competing binding sites are for the O₂, these effects may well shift binding energetics, especially for different binding motifs which should be addressed.

REVIEWER COMMENTS

Reviewer #1 (Remarks to the Author):

In this manuscript, the authors have explored the CO oxidation process on the
binuclear metal site of the heterogeneous Fe/MnO₂ catalyst. Sophisticated techniques
and theoretic calculations have been employed to characterize the site, analyze its
properties and explain the catalytic mechanism. It describes an interesting idea on
how to activate oxygen with such a binuclear site in the complex catalysis process.
However, the main conclusions are not convincing based on the provided
experimental evidence. Thus I could not recommend its acceptance in its current form.

1. The concept of “Binuclear single-atom” in the title is really confusing and
frustrating. It is common sense that each atom only has one nucleus. It seems that the
term actually means a two-Fe-atom site or bi-atomic or “binuclear iron” site in the
text. Why on earth do the authors use the “single-atom” concept?

2. The existence of binuclear iron sites is demonstrated with the HAADF-STEM
images, but unfortunately other sites like single or triple iron atoms cannot be
excluded. It’s hard to believe that all iron atoms maintain the binuclear configuration.
It seems that the manuscript selectively presents a few examples of neighboring
binuclear iron atoms. Even so, one can still find that there is at least a single iron atom
in Fig. S11 (4). Therefore the manuscript should provide the statistical distribution of
the sites as a function of iron atom number inside.

3. It’s not a good choice to directly compare the activities of Fe/MnO₂ and MnO₂
because typically a metal entity possesses a higher activity than an oxide support for
the CO oxidation no matter how the metal atoms are dispersed on the oxide.

4. There is not strong evidence to prove the existence of the Fe(O=O)Fe intermediate.
A new oxygen species is confirmed by O₂-TPD and H₂-TPR, but it could be any type
of active oxygen species. The O1s XPS in Fig. S8c simply indicates the oxide peak
and the wide feature of adsorbed oxygen is not any clue of surface-adsorbed O₂
species. Even if the oxygen adsorbs on the surface as O=O, there is no evidence to
show that it sits in between two iron atoms.

5. The 1204 cm⁻¹ band in Fig. 3f is supposed to be relating to the Fe(O=O)Fe species.
However, the authors should tell the readers that such a decrease in the wavenumber
is indeed from the isotopic effect for the Fe(O=O)Fe species. The Fe(O=O)Fe species
is non-polar, how could the O-O vibrational mode appear in IR? DFT could help
calculate all vibrational modes of the proposed Fe(O=O)Fe species. Such calculations
should be included.

6. An additional question about the blank experiment on the structure of the Fe/MnO₂

catalyst in Figure 1f: is there any oxygen molecule between the iron atoms at the
binuclear site before O₂ adsorption takes place? According to the analysis in the
manuscript, the active site should be Fe-(Ov)-Fe, then how can the O₂ molecule be
adsorbed to form Fe(O=O)Fe? The EXAFS result does show the existence of Fe-O
and Fe-(O)-Fe. Normally, Mn-O-Mn should be stable in the MnO₂ structure.
Therefore, Fe-O-Fe should be more stable than Fe-(Ov)-Fe and Fe(O=O)Fe, then why
is Fe-O-Fe non-existent in Fe/MnO₂?

Reviewer #2 (Remarks to the Author):

The authors claimed that they demonstrated that dual adjacent single Fe atoms
anchored on MnO₂ can assemble into a binuclear site, which activates molecular
oxygen to form an active intermediate species Fe(O=O)Fe for highly efficient CO
oxidation. It was concluded that binuclear Fe sites exhibited a stronger O₂ activation
performance than the conventional surface oxygen vacancy activation sites.

However, the data provided seems not enough to support their claims. The following
is the reasons why I consider it is inadequate, including questions and comments. I
hope this will help improve the quality of this author's work and contribute further
understanding the physics behind this seminal catalytic reaction.

In Fig.1a, TEM with atomic resolution are shown and Fe and Mn atoms were
identified by line profile in Fig. 3b. I assume that they were distinguished by the
intensity differences of the bright spots from the line profile. If so, I suggest the
authors to show TEM for MnO₂ to confirm that there is no intensity differences
among the bright spots and prove intensity difference is inherently from Fe and Mn in
Fe/MnO₂. I can see some bright spots on the surface of pure MnO₂ in HRTEM of Fig.
3a of the following literature, but their intensities are different.

72 J. Gao et al./Journal of Catalysis 341 (2016) 82-90.

I also suggest showing line profile for longer distance and calculating the Fe coverage
to see if it is consistent to the amount of introduced Fe. In some areas, three or more
bright spots appear to be lined up.

XAFS was taken for Fe foil (metallic Fe), Fe₂O₃ (Fe³⁺) and Fe/MnO₂ and spectrum of
Fe/MnO₂ is almost the same to that of Fe₂O₃. I suggest authors to discuss oxidation
state of Fe based on the result of XPS of Fe on Fe/MnO₂ with literatures and Bader
charge.

Fourier-transformed k-edge EXAFS for Fe₂O₃ and Fe/MnO₂ are quite similar. The
possibility of formation Fe₂O₃ was excluded by comparing the Morlet wavelet
transform in Supplementary Fig.3 but the results show no significant difference. The
k³-weighted EXAFS can be judged to check the quality of the measured spectra but
not shown. It may not be possible to prove that Fe₂O₃ is not formed only from the
results of this XAFS analysis.

Peaks in Fig 3a is too small to see.

How can you exclude the possibility of MvK on bare MnO₂ surface of Fe/MnO₂? Is
there any evidence that MvK is hindered by Fe atoms?

The Fe(O=O)Fe was suggested to the preferentially adsorbed configuration of O₂
since the adsorption energy is high, however, in the calculation in Supplementary
Fig. 10, Mn(O=O)Mn was not calculated.

What is the meaning of “molecular oxygen activation” and “activation behavior of
oxygen” in line 32 and 181? Is it mentioning low dissociation energy of oxygen
molecules? I suggest the authors define the term and share the physical image clearly
before it is used for the readership of this multidisciplinary journal.

In line 24, in introductory part, author described that high concentration of oxygen in
air or high temperature will inevitably lead to the refilling of oxygen vacancy by
referencing ORR study and not CO oxidation. I don't think this is appropriate citation.
There are many studies for oxygen vacancy involved CO oxidation studies. I suggest
authors to present discussions based on more relevant literatures. For the same
reasons, reference 1 to 3 seems inappropriate (they are electrochemical ORR studies
and there should be more relevant references).

Injection of Cl₂ was performed. Is it necessarily be Cl₂? Why Cl₂? I recommend
authors to add explanation why this way is good for investigating/identifying the CO
oxidation mechanism, probably with citing some articles.

In Supplementary Fig. 11 STEM images were shown but two of them are labeled the
same. I recommend that you carefully check the other data to make sure there are no
misrepresentations. Unit error was found in Line 108 and it should be 167 °C, not
116 cm⁻¹.

In Supplementary Fig. 6, fitted light off curve for MnO₂ is awkward. Fitting needs to
be performed based on the provided data and how you fit needs to be mentioned with
reliability factor.

Reviewer #3 (Remarks to the Author):

In this work, the authors report on a novel “single-atom” binuclear iron site embedded
in a MnO₂ host lattice. Overall, this work is of very high quality and the findings are
corroborated with theoretical study. This kind of earth-abundant material may well
serve as the next generation of catalyst for a variety of chemical technologies and this
type of fundamental study will be valuable in identifying paths forward in both
improving and utilizing such structures. A few minor suggestions that might improve
this manuscript:

1) While I understand the usage in comparison to previous work, the portion of the

title “Binuclear single-atom irons” is a bit problematic as the binuclear and plural of
iron both seem to contradict the “single-atom” term. This term is of course somewhat
problematic for all “single-atom” sites since local structure can strongly influence key
binding sites but even more so here where 2 distinct atoms are serving as the localized
active site. A similar issue occurs at the end of the discussion on page 12 where the
term “monatomic binuclear active sites” is used.

2) Oxygen activation is of course key to a wide variety of (electro)chemical reactions.
A sentence on some of the most technologically important applications and some
additional discussion on the importance of CO \diamond CO₂ in particular would help
highlight the importance of this work.

3) On a technical level, the denticity/binding motif of the bound O₂ as brought up on
page 7 and Figure S10 could be strengthened somewhat. In particular, it seems odd to
not have considered and reported on the energetics of the O₂ with 1 O bonded to each
of the Fe atoms, even if it is indeed higher energy than the O₂ bridging the 2 Fe with a
single O. Also, based on equation 5, it appears that binding does not include
vibrational contributions to the free energy including zero-point energies. Given how
close some of the competing binding sites are for the O₂, these effects may well shift
binding energetics, especially for different binding motifs which should be addressed.

Response Letter

We thank all three Reviewers for their positive reviews of this manuscripts and the
constructive suggestions that help to improve the scientific presentation of this
manuscript. Point-by-Point responses to address the concerns raised by the three
Reviewers are shown in the following.

**Reviewer #1 (Remarks to the Author):**

*In this manuscript, the authors have explored the CO oxidation process on the*
*binuclear metal site of the heterogeneous Fe/MnO₂ catalyst. Sophisticated techniques*
*and theoretic calculations have been employed to characterize the site, analyze its*
*properties and explain the catalytic mechanism. It describes an interesting idea on*
*how to activate oxygen with such a binuclear site in the complex catalysis process.*
*However, the main conclusions are not convincing based on the provided*
*experimental evidence. Thus, I could not recommend its acceptance in its current*
*form.*

We thank the Reviewer for confirming the important significance of our findings and
for approving our extensive catalysis studies, and for making many insightful
comments. During the revision, we provided more experimental evidences to support
our conclusions.

*1. The concept of “Binuclear single-atom” in the title is really confusing and*
*frustrating. It is common sense that each atom only has one nucleus. It seems that the*
*term actually means a two-Fe-atom site or bi-atomic or “binuclear iron” site in the*
*text. Why on earth do the authors use the “single-atom” concept?*

**Response:** We thank the reviewer for the insightful comments. To avoid the confusion,
we revised the title into *Adjacent single-atom irons boosting molecular oxygen*
*activation on MnO₂* and also corrected the relevant term thoroughly in the paper. The
purpose for us to use the concept of single atom is to emphasize the importance of
surrounding environment of iron atoms and their interaction with MnO₂ support on

the performance of molecular oxygen activation.

*2. The existence of binuclear iron sites is demonstrated with the HAADF-STEM*
*images, but unfortunately other sites like single or triple iron atoms cannot be*
*excluded. It's hard to believe that all iron atoms maintain the binuclear configuration.*
*It seems that the manuscript selectively presents a few examples of neighboring*
*binuclear iron atoms. Even so, one can still find that there is at least a single iron*
*atom in Fig. S11 (4). Therefore, the manuscript should provide the statistical*
*distribution of the sites as a function of iron atom number inside.*

*Response: We thank the Reviewer for the great suggestion. As shown in Figure S15*
*(Figure S11 in previous submission), single Fe sites were unavoidable during the*
*formation of adjacent iron sites. Therefore, based on the strength of atoms and the*
*distance between adjacent atoms, we statistically analyzed the STEM spectra of 300*
*atoms in three regions on the surface of three different Fe/MnO₂ catalysts (0.1%, 0.25%*
*and 0.5% Fe/MnO₂) in Figure R1 and Table R1. For 0.25% Fe/MnO₂ shown in the*
*Figure R1b, 1e and Table R1, 80.9% of Fe sites (n = 2 and n > 2) (n represents the*
*number of adjacent Fe atoms) on the surface were distributed as adjacent Fe sites, and*
*a small number (19.1%) as monatomic Fe site. Obviously, these adjacent Fe sites*
*strongly contributed to the efficient activation of molecular oxygen.*

*During the revision, Figure R1 and Table R1 were added as the new Supplementary*
*Figure S16 and Table S2 respectively. Meanwhile, we also added more discussions in*
*the revised manuscript on page 8 about the statistical distribution of the sites.*

Figure R1. (a-c) STEM images of different Fe/MnO₂ catalysts. The atoms surrounded
 by the blue dotted box are used for site statistics. (d-f) Statistical results of the number
 of Fe sites corresponding to different catalysts.

Table R1. Statistical results of different Fe sites and Mn contents of the three catalysts
 from Figure R1.

	Fe(total)	Fe(n=1)	Fe(n=2)	Fe(n>2)	Mn
0.1% Fe/MnO ₂	43	35	8	0	257
0.25% Fe/MnO ₂	89	17	67	5	211
0.5% Fe/MnO ₂	172	12	34	126	128

*n* represents the number of adjacent Fe atoms.

3. It's not a good choice to directly compare the activities of Fe/MnO₂ and MnO₂
 because typically a metal entity possesses a higher activity than an oxide support for
 the CO oxidation no matter how the metal atoms are dispersed on the oxide.

Response: We thank the reviewer for this valuable comment. As the reviewer
 mentioned, a metal usually possesses a higher activity than an oxide support in CO
 oxidation reaction. Therefore, we compared the activities of catalysts with different Fe
 contents (0.1%, 0.25% and 0.5% Fe/MnO₂), and found that 0.25% Fe/MnO₂ exhibited
 the highest catalytic activity (Figure R2). According to the statistical analysis of iron

atoms distribution on the surface of three catalysts (Figure R1 and Table R1), 0.1%
Fe/MnO₂ possessed 81.4% of monatomic Fe, while 0.25% Fe/MnO₂ was of 75.3%
two adjacent Fe sites. When the content of Fe reached 0.5%, the adjacent Fe sites ($n >$
2) became dominant (73.2%). Obviously, the catalytic activity of Fe/MnO₂ strongly
depends on the surrounding environment of iron atoms and their interaction with
MnO₂ support.

*During the revision, we added the corresponding discussion on page 17 in the revised*
*Supporting Information as follows. “We compared the activities of catalysts with*
*different Fe introduction (0.1%, 0.25% and 0.5% Fe/MnO₂), and found that 0.25%*
*Fe/MnO₂ exhibited the highest catalytic activity (Figure S7a). According to the*
*statistical analysis of iron atoms distribution on the surface of three catalysts*
*(Figure S16 and Table S2), 0.1% Fe/MnO₂ possessed 81.4% of monatomic Fe, while*
*0.25% Fe/MnO₂ was of 75.3% two adjacent single-atom Fe sites. When the content*
*of Fe reached 0.5%, the adjacent Fe sites ($n > 2$) became dominant (73.2%).*
*Obviously, the catalytic activity of Fe/MnO₂ strongly depends on the surrounding*
*environment of iron atoms and their interaction with MnO₂ support.”*

**Figure R2.** Light-off curves for CO oxidation of different proportions of Fe/MnO₂.

4. There is not strong evidence to prove the existence of the Fe(O=O)Fe intermediate.

A new oxygen species is confirmed by O₂-TPD and H₂-TPR, but it could be any type

*of active oxygen species. The O1s XPS in Fig. S8c simply indicates the oxide peak and*
*the wide feature of adsorbed oxygen is not any clue of surface-adsorbed O₂ species.*
*Even if the oxygen adsorbs on the surface as O=O, there is no evidence to show that it*
*sits in between two iron atoms.*

*Response: Thanks a lot for the comment. During the revision, we employed in-situ*
*DRIFTS with ¹⁸O₂ isotope to further confirm the existence of the Fe(O=O)Fe*
*intermediate on the surface of Fe/MnO₂, and found that the characteristic peak of*
*1204 cm⁻¹ shifted to the direction of low wavenumber owing to isotopic effect (Figure*
*R3), suggesting this peak was associated with O₂. As for MnO₂, the in-situ DRIFTS*
*with ¹⁶O₂ did not display any peaks near 1204 cm⁻¹ (Figure R4). Accordingly, the*
*adsorption site of O₂ was related to Fe species. In light of the calculation results*
*(Figure R5) that O₂ was adsorbed on the two adjacent Fe sites (-2.00 eV) with the*
*highest adsorption energy, the Fe(O=O)Fe was the most favorable adsorption*
*configuration. Furthermore, we calculated the infrared spectra of Fe/MnO₂ and*
*Fe/MnO₂ with O₂ adsorption in the configuration of Fe(O=O)Fe (Figure R6). In the*
*simulated infrared spectrum of Fe/MnO₂ with O₂ adsorption, a new characteristic*
*peak appeared near 1200 cm⁻¹. We also performed the EXAFS fitting of Fe(O=O)Fe,*
*which were highly coincident with the experimental results (Figure R7). Therefore,*
*we believed that O₂ was adsorbed to the two adjacent Fe sites with the binuclear*
*configuration and was thus activated.*

*During the revision, Figure R5 was modified as the new Supplementary Figure S13,*
*Figure R6 was added as a new Figure 3c in the revised manuscript and the*
*corresponding discussion was added on page 8 in the revised manuscript as follows.*

*“Furthermore, we simulated the infrared spectra of Fe/MnO₂ and Fe/MnO₂ with*
*O₂ adsorption by DFT calculation (Figure 3c). In the simulated infrared spectrum*
*of Fe/MnO₂ with O₂ adsorption, a new characteristic peak appeared near 1200 cm⁻¹,*
*which was consistent with the experiment results.”*

Figure R3. In situ DRIFTS of ¹⁸O₂ isotope over Fe/MnO₂ in the out-line system.

Figure R4. In situ DRIFTS of ¹⁶O₂ over MnO₂ in the out-line system.

Configuration	a	b	c	d	e	f	g	h	i
Adsorption Energy (eV)	-2.00	-1.47	-1.82	-0.83	-1.66	-2.15	-3.11	-2.39	-1.69

Figure R5. The stable configurations of (a-e) O₂, (f) CO and (g) Cl₂ adsorbed at the
 different sites of Fe/MnO₂ surface, and O₂ adsorbed in (h) the oxygen vacancy or (i)
 bi-manganese sites of MnO₂. The corresponding adsorption energy with zero-point
 energy correction is listed in the table of the picture.

Figure R6. Simulated infrared spectra of Fe/MnO₂ and Fe/MnO₂ with O₂ adsorption.

The bottom of the picture was the structure of the different species.

Figure R7. (a) Fourier-transformed K-edge EXAFS spectra in R-space of Fe (without
phase correction) and the fitting of Fe/MnO₂ structure in (b).

5. The 1204 cm⁻¹ band in Fig. 2f is supposed to be relating to the Fe(O=O)Fe species.

However, the authors should tell the readers that such a decrease in the wavenumber

is indeed from the isotopic effect for the Fe(O=O)Fe species. The Fe(O=O)Fe species

is non-polar, how could the O-O vibrational mode appear in IR? DFT could help

calculate all vibrational modes of the proposed Fe(O=O)Fe species. Such

calculations should be included.

Response: We thank the reviewer for this valuable advice. The simulated infrared
 spectra (Figure R6) confirmed that the 1204 cm^{-1} band in Figure 2f was related to the
 $\text{Fe}(\text{O}=\text{O})\text{Fe}$ species. Previously studies revealed that the decrease in the wavenumber
 of in-situ DRIFTS with $^{18}\text{O}_2$ was arisen from the isotopic effect [please see *Nat. Catal.*
 *2*, 916-924 (2019)], which was added on page 7 in the revised manuscript as follows.
 *“When $^{16}\text{O}_2$ was replaced by $^{18}\text{O}_2$, this characteristic peak shifted from 1204 cm^{-1} to*
 *the lower wavenumber of 1159 cm^{-1} due to the isotopic effect (Figure 3a), indicating*
 *that this peak was closely related to the new active oxygen species generated via O_2*
 *activation on Fe/MnO_2 [22].”*

Bader charge calculation results revealed the two oxygen atoms had different
 charges (-0.37 and -0.06) after an oxygen molecule was adsorbed on the two adjacent
 Fe sites with end-on mode (Figure R8). The different charges constantly changed the
 dipole moment of O_2 in the vibration process, accounting for the infrared
 characteristic absorption peak of $\text{Fe}(\text{O}=\text{O})\text{Fe}$. During the revision, we simulated
 infrared spectra of Fe/MnO_2 and Fe/MnO_2 with O_2 adsorption by DFT calculation. As
 expected, a new characteristic peak appeared near 1200 cm^{-1} for Fe/MnO_2 with O_2
 adsorption (Figure R6), consistent with the experimental results.

Figure R8. The difference charge density of O_2 adsorbed on Fe/MnO_2 . Different

oxygen atoms from adsorbed oxygen species were labeled as a and b, and
corresponding Bader charge was recorded in the bottom of charge density map. The
charge density of yellow and blue represents the concentrated and scarce electrostatic
potential scale respectively.

*6. An additional question about the blank experiment on the structure of the Fe/MnO₂*
*catalyst in Figure 1f: is there any oxygen molecule between the iron atoms at the*
*binuclear site before O₂ adsorption takes place? According to the analysis in the*
*manuscript, the active site should be Fe-(O_v)-Fe, then how can the O₂ molecule be*
*adsorbed to form Fe(O=O)Fe? The EXAFS result does show the existence of Fe-O*
*and Fe-(O)-Fe. Normally, Mn-O-Mn should be stable in the MnO₂ structure.*
*Therefore, Fe-O-Fe should be more stable than Fe-(O_v)-Fe and Fe(O=O)Fe, then*
*why is Fe-O-Fe non-existent in Fe/MnO₂?*

**Response:** We thank the reviewer for the valuable comments. We would like to
answer the comments as follows.

(1) The DFT calculation results demonstrated that O₂ was adsorbed on the two
adjacent Fe sites (-2.00 eV) with the highest adsorption energy, indicating that the
Fe(O=O)Fe was the most favorable adsorption configuration (Figure R5a). Regarding
that the catalyst after hydrothermal reaction was calcined in air atmosphere during the
preparation process of Fe/MnO₂, we believe that oxygen molecules might be adsorbed
between the two adjacent Fe sites before O₂ adsorption.

(2) The formation of Fe(O=O)Fe might be related to the synthesis method of the
catalyst. To verify this assumption, we conducted ATR measurements of Fe/MnO₂
synthesized at different temperatures (Figure R9). The characteristic peaks of ferric
oxalate (757, 812 and 1255 cm⁻¹) appeared in the catalysts synthesized at 80 °C,
which were weakened along with the further increase of the synthesis temperatures. In
combination with Figure S25 and S26, the iron species existed in the precursor
solution with the form of iron oxalate complex. With the increase of hydrothermal
temperature, [MnO₆] structural units appeared in priority, resulting in the formation of
the periodic structure of MnO₂. Then, the ferric oxalate was gradually decomposed

into CO₂, leaving Fe atoms on the surface of MnO₂. Regarding that oxalate was
excessive and the reaction was in a reductive environment [please see *J. Hazard.*
*Mater.*, 262, 701-708 (2013) and *Environ. Sci. Technol.*, 53, 6444-6453 (2019)],
oxygen vacancies were more easily generated on the catalyst surface. Therefore, we
believe that the two adjacent sites possessed the Fe-(O_v)-Fe configuration and thus
formed Fe(O=O)Fe species after calcination in the air atmosphere,.

(3) We also simulated the infrared spectrum of Fe/MnO₂ with Fe-O-Fe structure, and
did not find the appearance of characteristic peak near 1200 cm⁻¹ (Figure R10). Thus,
more active Fe(O=O)Fe can be obtained through the hydrothermal method of
oxalate-assisted chelation coordination, even though Fe-O-Fe is more stable.

*During the revision, Figure R9 was added as new Supplementary Figure S24.*

Figure R9. ATR spectra of standard Fe₂(C₂O₄)₃ and Fe/MnO₂ synthesized from at
different temperatures.

Figure R10. Simulated infrared spectrum of Fe/MnO₂ with Fe-O-Fe structure. The

bottom of the picture was the structure of the different species

**Reviewer #2 (Remarks to the Author):**

*The authors claimed that they demonstrated that dual adjacent single Fe atoms*
*anchored on MnO₂ can assemble into a binuclear site, which activates molecular*
*oxygen to form an active intermediate species Fe(O=O)Fe for highly efficient CO*
*oxidation. It was concluded that binuclear Fe sites exhibited a stronger O₂ activation*
*performance than the conventional surface oxygen vacancy activation sites.*

*However, the data provided seems not enough to support their claims. The following*
*is the reasons why I consider it is inadequate, including questions and comments. I*
*hope this will help improve the quality of this author's work and contribute further*
*understanding the physics behind this seminal catalytic reaction.*

Thank you for your summary. We appreciate your efforts in reviewing our manuscript
and have revised the manuscript accordingly to support our conclusions as strongly as
possible.

*In Fig.1a, TEM with atomic resolution are shown and Fe and Mn atoms were*
*identified by line profile in Fig. 3b. I assume that they were distinguished by the*
*intensity differences of the bright spots from the line profile. If so, I suggest the*
*authors to show TEM for MnO₂ to confirm that there is no intensity differences among*
*the bright spots and prove intensity difference is inherently from Fe and Mn in*
*Fe/MnO₂. I can see some bright spots on the surface of pure MnO₂ in HRTEM of Fig.*
*3a of the following literature, but their intensities are different.*

*J. Gao et al./Journal of Catalysis 341 (2016) 82-90.*

*I also suggest showing line profile for longer distance and calculating the Fe*
*coverage to see if it is consistent to the amount of introduced Fe. In some areas, three*
*or more bright spots appear to be lined up.*

Response: We thank the reviewer for this constructive advice and would like to reply
the comments as follows.

(1) We distinguished Mn and Fe atoms by the intensity difference and the distance
between the atoms from HAADF-STEM, because the intensity of Fe is stronger than
that of Mn. Meanwhile, DFT calculation results revealed the distance between Fe-Fe

was 3.1 Å, longer than that 2.7 Å of Fe-Mn. Referring to the remarkable article [*J.*
 *Catal.*, 341, 82-90 (2016)], we performed STEM test on pure MnO₂ (Figure R11). The
 results revealed that the intensity and the distance of 2.9 Å between the atoms were in
 accordance with the theoretical calculation model of MnO₂.

(2) Based on the strength of atoms and the distance between adjacent atoms, we
 statistically analyzed the STEM spectra of 300 atoms in three regions on the surface
 of three different Fe/MnO₂ catalysts (0.1%, 0.25% and 0.5% Fe/MnO₂) in Figure R1
 and Table R1. For 0.25% Fe/MnO₂, 89 of 300 atoms on the surface were counted as
 Fe atoms, and others as Mn atoms. According to the model structure of Fe/MnO₂
 (Figure R12), the height of the monolayer atoms is 2.63 Å (h). The TEM results
 displayed that the thickness of the nanorods was about 19.7 nm (H), consistent with
 the literature [*ACS Catal.* 8, 3435-3446 (2018)]. On the basis of calculation formulas
 (1-5), the mass fraction of Fe was 0.46%, very close to the ICP test results (0.3%), if
 the impurities adsorbed on the surface were ignored. These results have demonstrated
 that Fe is distributed as a single atom on the surface of Fe/MnO₂, and it is feasible to
 distinguish Mn and Fe atoms according to the differences in strength and distance.

$$413 \quad m_{\text{Fe}} = \frac{n_{\text{Fe}} \times M_{\text{Fe}}}{N_{\text{A}}} \quad (1)$$

$$414 \quad m_{\text{Mn}} = \frac{n_{\text{Mn}} \times M_{\text{Mn}}}{N_{\text{A}}} \quad (2)$$

$$415 \quad m_{\text{O}} = \frac{2 \times (n_{\text{Mn}} + n_{\text{Fe}}) \times M_{\text{O}}}{N_{\text{A}}} \quad (3)$$

$$416 \quad N = \frac{H}{h} \quad (4)$$

$$417 \quad W_{\text{Fe}} = \frac{2 \times m_{\text{Fe}}}{2 \times (m_{\text{Fe}} + m_{\text{Mn}} + m_{\text{O}}) + 300 \times m_{\text{MnO}_2} \times N} \quad (5)$$

Where m represents the mass of different element; N is the number of atoms; M is the
 relative atomic mass; N_A is the Avogadro constant; H is the thickness of catalyst,
 which is counted by TEM test results. h represents the height of single-layer MnO₂,
 which comes from the structural optimization model; N is the number of layers; W_{Fe}

is the mass fraction of Fe;

*During the revision, Figure R11 and R12 were added as the new Supplementary*
*Figure S14 and S17 respectively. We also cited this relevant literature [J. Catal., 341*
*82-90 (2016)] as Ref. 23 and added the corresponding discussion on page 18 in the*
*revised Supporting Information as follows. “Based on the strength of atoms and the*
*distance between adjacent atoms, we statistically analyzed the STEM spectra of 300*
*atoms in three regions on the surface of three different Fe/MnO₂ catalysts (0.1%,*
*0.25% and 0.5% Fe/MnO₂) in Figure S16 and Table R2. For 0.25% Fe/MnO₂, 89 of*
*300 atoms on the surface were counted as Fe atoms, and others as Mn atoms.*
*According to the model structure of Fe/MnO₂ (Figure S17), the height of the*
*monolayer atoms is 2.63 Å (h). The TEM results displayed that the thickness of the*
*nanorods was about 19.7 nm (H), consistent with the literature [4]. On the basis of*
*calculation formulas (5-9), the mass fraction of Fe was 0.46%, very close to the ICP*
*test results (0.3%), if the impurities adsorbed on the surface were ignored. These*
*results have demonstrated that Fe is distributed as a single atom on the surface of*
*Fe/MnO₂, and it is feasible to distinguish Mn and Fe atoms according to the*
*differences in strength and distance.”*

 Figure R11. STEM image of MnO₂ (top), intensity surface plot from blue dashed
 rectangle (middle) and the corresponding structural model (bottom).

 Figure R1. (a-c) STEM images of different Fe/MnO₂ catalysts. The atoms surrounded
 by the blue dotted box will be used for site statistics. (d-f) Statistical results of the

number of Fe sites corresponding to different catalysts.

Table R1. Statistical results of different Fe sites and Mn contents of the three catalysts

from Figure R1.

	Fe(total)	Fe(n=1)	Fe(n=2)	Fe(n>2)	Mn
0.1% Fe/MnO ₂	43	35	8	0	257
0.25% Fe/MnO ₂	89	17	67	5	211
0.5% Fe/MnO ₂	172	12	34	126	128

n represents the number of adjacent Fe atoms.

Figure R12. (a) The structural model and (b) TEM spectrum of Fe/MnO₂. Particle size
was counted according to TEM result.

XAFS was taken for Fe foil (metallic Fe), Fe₂O₃ (Fe³⁺) and Fe/MnO₂ and spectrum of
Fe/MnO₂ is almost the same to that of Fe₂O₃. I suggest authors to discuss oxidation
state of Fe based on the result of XPS of Fe on Fe/MnO₂ with literatures and Bader
charge.

Fourier-transformed k-edge EXAFS for Fe₂O₃ and Fe/MnO₂ are quite similar. The
possibility of formation Fe₂O₃ was excluded by comparing the Morlet wavelet
transform in Supplementary Fig.3 but the results show no significant difference. The
k³-weighted EXAFS can be judged to check the quality of the measured spectra but
not shown. It may not be possible to prove that Fe₂O₃ is not formed only from the

*results of this XAFS analysis.*

Response: Thank you for your kind comments and helpful suggestions. We used
ammonium oxalate and iron nitrate to synthesize Fe_2O_3 , and measured the Fe 3d XPS
spectra of pure Fe_2O_3 and Fe/MnO_2 (Figure R13). It was found that the characteristic
peak of Fe/MnO_2 had lower binding energy (710.6 eV) than that (711.4 eV) of Fe_2O_3 .
Meanwhile, Bader charge calculation results revealed that Fe on Fe/MnO_2 possessed
more electrons than that of Fe_2O_3 , consistent with the results of XANES absorption
(Figure R14). Furthermore, the statistical results of Fe atoms distributed on the
surface of 0.25% Fe/MnO_2 from STEM spectrum (0.46%) matched with the result
from ICP test (0.3%). These results ruled out the formation of Fe_2O_3 in the Fe/MnO_2 .
k3-weighted EXAFS was shown in Fig. R15 (Figure S3 of revised Supporting
Information).

*During the revision, Figure R13 and R15 were added as new Figure S6 and Figure S3*
*in the revised Supporting Information respectively, and more discussion was added on*
*page 5 in the manuscript as follows. “We also compared the Fe 3d XPS spectra of*
*Fe_2O_3 and Fe/MnO_2 (Figure S6), and found that the characteristic peak of*
*Fe/MnO_2 had lower binding energy (710.6 eV) than that (711.4 eV) of Fe_2O_3 .*
*Meanwhile, Bader charge calculation results revealed that Fe on Fe/MnO_2*
*possessed more electrons than that of Fe_2O_3 , consistent with the results of XANES*
*absorption (Figure 1d).”*

Figure R13. XPS spectrum of Fe2p in Fe/MnO₂ and Fe₂O₃ sample.

Figure R14. Normalized XANES spectra of Fe.

Figure R15. k3-weighted EXAFS spectrum of Fe/MnO₂ sample.

*Peaks in Fig 3a is too small to see.*

*Response: Thanks for the suggestions. We enlarged the peaks in Figure R3 for better*
 *view (Figure 3a in the revised manuscript).*

Figure R3. In situ DRIFTS of ¹⁸O₂ isotope over Fe/MnO₂ in the out-line system.

*How can you exclude the possibility of MvK on bare MnO₂ surface of Fe/MnO₂? Is*

*there any evidence that MvK is hindered by Fe atoms?*

*The Fe(O=O)Fe was suggested to the preferentially adsorbed configuration of O₂*
*since the adsorption energy is high, however, in the calculation in Supplementary*
*Fig .10, Mn(O=O)Mn was not calculated.*

*Response: We thank the reviewer for these valuable questions and would like to reply*
*the comments as follows.*

(1) We adopted DFT calculation to check why the MvK mechanism on MnO₂ surface
was hindered by Fe atoms. First of all, O₂ located at the unsaturated Fe sites possessed
a higher energy (-2.00 eV in Figure R5a) than that at the oxygen vacancy (-0.83 eV in
Figure R5d). Then, the calculation results of the transition state revealed that the
energy (0.19 eV) for CO reacting with O₂ adsorbed on the two adjacent Fe sites was
lower than that (0.37 eV) of O₂ adsorbed in the oxygen vacancy (Figure R16).
Therefore, the MvK mechanism on bare MnO₂ was inhibited by the high energy
demand.

(2) In accordance with the reviewer's suggestion, we calculated the oxygen adsorption
energy (-1.69 eV) of Mn(O=O)Mn configuration (Figure R5i), which was lower than
that (-2.00 eV) of Fe(O=O)Fe.

*During the revision, Figure R16 was added as the new Supplementary Figure S29 and*
*the corresponding discussion was added on page 13 in the revised manuscript as*
*follows. "Furthermore, we also calculated the energy of CO reacting with O₂ on*
*different sites. The calculation results of the transition state revealed that the*
*energy (0.19 eV) for CO reacting with O₂ adsorbed on the adjacent Fe sites was*
*lower than that (0.37 eV) of O₂ adsorbed on the V_o referring to Figure S29.*
*Therefore, the MvK mechanism on bare MnO₂ could be inhibited by the high*
*energy demand."*

Configuration	a	b	c	d	e	f	g	h	i
Adsorption Energy (eV)	-2.00	-1.47	-1.82	-0.83	-1.66	-2.15	-3.11	-2.39	-1.69

Figure R5. The stable configurations of (a-e) O₂, (f) CO and (g) Cl₂ adsorbed at the
 different sites of Fe/MnO₂ surface, and O₂ adsorbed in (h) the oxygen vacancy or (i)
 bi-manganese sites of MnO₂. The corresponding adsorption energy with zero-point
 energy correction is listed in the table of the picture.

Figure R16. Energy profiles of CO reacted through the different path in Fe/MnO₂. The
 optimized structures of initial states (I), transition states (TS) and final states (II) are
 listed in the dotted box, and the relative energy is recorded below the corresponding
 structure.

*What is the meaning of “molecular oxygen activation” and “activation behavior of*
 *oxygen” in line 32 and 184? Is it mentioning low dissociation energy of oxygen*
 *molecules? I suggest the authors define the term and share the physical image clearly*
 *before it is used for the readership of this multidisciplinary journal.*

**Response:** We appreciate the reviewer for these kind suggestions. In the original
 manuscript, both molecular oxygen activation and activation behavior of oxygen
 corresponded to weakening O=O double bond, reducing dissociation energy of
 oxygen molecule and facilitating oxygen to participate in the reaction process more
 easily. The diagram of molecular oxygen activation was shown in Figure R17. We
 added more description for better understanding of this term.

*During the revision, Figure R17 was added as the new Supplementary Figure S1 and*
 *the corresponding discussion about the term of molecular oxygen activation was*
 *added on page 2 in the revised manuscript and page 2 in the Supporting Information*
 *as follows. “Molecular oxygen activation is a continuous process of adsorption and*
 *dissociation of O₂ on the catalyst surface. The process relies on the transfer of*

*electrons from the surface of catalyst to the O₂ to weaken the oxygen-oxygen double*
*bond. Meanwhile, the structure of the adsorption site on the catalyst surface plays*
*an important role in the activation of O₂. [Chem. Rev., 118, 2816-2862 (2018); Nat*
*Commun. 12, 2741 (2021)]”*

Figure R17. Diagram of molecular oxygen activation.

*In line 24, in introductory part, author described that high concentration of oxygen in*
*air or high temperature will inevitably lead to the refilling of oxygen vacancy by*
*referencing ORR study and not CO oxidation. I don't think this is appropriate citation.*

*There are many studies for oxygen vacancy involved CO oxidation studies. I suggest*
*authors to present discussions based on more relevant literatures. For the same*
*reasons, reference 1 to 3 seems inappropriate (they are electrochemical ORR studies*
*and there should be more relevant references).*

*Response: We thank the reviewer for the valuable suggestion. We deleted references*
*1-4 from the original manuscript and replaced them with new references 7-9 [J. Am.*
*Chem. Soc. 140, 4580-4587 (2018); ACS Catal. 9, 9751-9763 (2019); ChemCatChem*
*9, 1119-1127 (2017)]. These references offer us deeper atomic-level insights into the*
*relationship between oxygen vacancy and molecular oxygen activation, promoting*
*CO oxidation.*

*Injection of Cl₂ was performed. Is it necessarily be Cl₂? Why Cl₂? I recommend*
*authors to add explanation why this way is good for investigating/identifying the CO*
*oxidation mechanism, probably with citing some articles.*

*Response: We thank the reviewer for these valuable comments. Chlorine is well*
*known for its high electron density and strong adsorption properties [J. Am. Chem.*
*Soc., 2012, 134, 20160–20168]. Some literatures reported that chlorine was prone to*
*form strong adsorption with metal sites on the catalyst surface during the catalytic*
*reaction, leading to catalyst deactivation [please see Chem. Rev., 2019, 119,*
*4471–4568; Environ. Sci. Technol., 2021, 55, 4007–4016]. Even after oxygen*
*adsorption, Cl₂ could cover the surface of MnO₂ and firmly occupied the adsorption*
*sites of CO. Furthermore, Cl₂ has a stronger adsorption capacity at unsaturated Mn*
*sites than CO (-3.11 eV for Cl₂ versus -2.15 eV for CO), accounting for the design*
*idea of the experiment. Therefore, we used Cl₂ to study the reaction mechanism in this*
*study.*

*During the revision, we cited the related works as Refs. 26-28 and added more*
*discussion on page 22 in the Supporting Information as follows. “Chlorine is well*
*known for its high electron density and strong adsorption properties. Some*
*literatures reported that chlorine was prone to form strong adsorption with metal*
*sites on the catalyst surface during the catalytic reaction, leading to catalyst*
*deactivation [5, 6]. Even after oxygen adsorption, Cl₂ could cover the surface of*
*MnO₂ and firmly occupied the adsorption sites of CO. Furthermore, Cl₂ has a*
*stronger adsorption capacity at unsaturated Mn sites than CO (-3.11 eV for Cl₂*
*versus -2.15 eV for CO), accounting for the design idea of the experiment.*
*Therefore, we used Cl₂ to study the reaction mechanism in this study.”*

*In Supplementary Fig. 11 STEM images were shown but two of them are labeled the*
*same. I recommend that you carefully check the other data to make sure there are no*
*misrepresentations. Unit error was found in Line 108 and it should be 167 °C, not*
*cm⁻¹.*

Response: We are sorry for the typo error and carefully checked the manuscript
thoroughly.

*In Supplementary Fig. 6, fitted light off curve for MnO₂ is awkward. Fitting needs to*
*be performed based on the provided data and how you fit needs to be mentioned with*
*reliability factor.*

Response: We appreciate the reviewer for the valuable comment. Figure R18
corresponds to the activity spectrum after the normalization of specific surface area.
The relation between $C/(C_0 \cdot S)$ and T was used to plot the spectrum, rather than the
result of kinetic curve fitting.

Figure R18. (a) Light-off curves of CO oxidation over Fe/MnO₂ and MnO₂. Nitrogen
adsorption/desorption isotherm plots of (b) MnO₂ and (c) Fe/MnO₂ (the insets of a
and b showed the pore size distribution). (d) CO catalytic performance of the MnO₂
and Fe/MnO₂ normalized by BET surface area.

**Reviewer #3 (Remarks to the Author):**

*In this work, the authors report on a novel “single-atom” binuclear iron site*
*embedded in a MnO₂ host lattice. Overall, this work is of very high quality and the*
*findings are corroborated with theoretical study. This kind of earth-abundant material*
*may well serve as the next generation of catalyst for a variety of chemical*
*technologies and this type of fundamental study will be valuable in identifying paths*
*forward in both improving and utilizing such structures. A few minor suggestions that*
*might improve this manuscript:*

We thank the reviewer for the valuable and constructive comments and carefully
revised the manuscript.

1) While I understand the usage in comparison to previous work, the portion of the
title “Binuclear single-atom irons” is a bit problematic as the binuclear and plural of
iron both seem to contradict the “single-atom” term. This term is of course somewhat
problematic for all “single-atom” sites since local structure can strongly influence
key binding sites but even more so here where 2 distinct atoms are serving as the
localized active site. A similar issue occurs at the end of the discussion on page 12
where the term “monatomic binuclear active sites” is used.

Response: Thank you for your helpful comments. We agree with the reviewer’s
suggestion. To avoid the confusion, we revised the title into *Adjacent single-atom*
*irons boosting molecular oxygen activation on MnO₂* and also corrected the relevant
term thoroughly in the paper.

2) Oxygen activation is of course key to a wide variety of (electro)chemical reactions.
A sentence on some of the most technologically important applications and some
additional discussion on the importance of CO → CO₂ in particular would help
highlight the importance of this work.

Response: Thank you for your suggestion. In the introduction, we added the
discussion on the importance of CO → CO₂ to highlight the significance of this work
as follows. *“Molecular oxygen activation, a continuous process of adsorption and*

*dissociation of O₂ on the catalyst surface (Figure S1), is a key step in catalytic*
*reactions [Chem. Rev., 118, 2816-2862 (2018)], including the synthesis of organic*
*compounds, catalytic combustion of volatile organic compounds (VOCs), oxygen*
*reduction reaction (ORR) in fuel cells and so on [J. Am. Chem. Soc. 134,*
*13018-13026 (2012); ACS Catal. 11, 6614-6625 (2021); Nat. Catal. 4, 463-468*
*(2021)].”*

*“High-efficiency CO oxidation is of great significance in automotive exhaust*
*purification and the anti-toxicity improvement of proton exchange membrane fuel*
*cells [Science 358, 1419–1423 (2017); ACS Catal., 10, 6532–6545 (2020)].”*

*During the revision, more discussion about oxygen activation was added on page 2*
*and 5 in the revised manuscript and the related Refs. 1-4, 16 and 17 were cited.*

*3) On a technical level, the denticity/binding motif of the bound O₂ as brought up on*
*page 7 and Figure S10 could be strengthened somewhat. In particular, it seems odd to*
*not have considered and reported on the energetics of the O₂ with 1 O bonded to each*
*of the Fe atoms, even if it is indeed higher energy than the O₂ bridging the 2 Fe with a*
*single O. Also, based on equation 5, it appears that binding does not include*
*vibrational contributions to the free energy including zero-point energies. Given how*
*close some of the competing binding sites are for the O₂, these effects may well shift*
*binding energetics, especially for different binding motifs which should be addressed.*

*Response: We appreciate the insightful comments from the reviewer. During the*
*revision, we calculated the situation for each oxygen atom from O₂ attached to Fe site*
*(side-on configuration) (Figure R5b), and found that the adsorption energy (-1.47 eV)*
*of O₂ in side-on configuration was significantly lower than that (-2.00 eV) of the*
*end-on adsorption configuration (Fe(O=O)Fe) (Figure R5a). Thereby, the adsorption*
*of O₂ on Fe sites preferred to the end-on mode. Meanwhile, we also calculated the*
*Bader charge and the differential charge density of side-on configuration (Figure R19),*
*and found that two oxygen atoms got the same number of electrons, and the electron*
*density was evenly distributed among the adsorbed oxygen molecule. Therefore, the*
*dipole moment of oxygen molecules did not change during the vibration process,*

inconsistent with the results of the in-situ DRIFTS. As a result, the adsorption
 configuration of oxygen on Fe sites in side-on mode was further excluded.

Zero-point energy correction (Table R2) was carried out for each adsorption
 configuration, and the corresponding adsorption energies was recalculated (Figure
 R5).

*During the revision, Figure R19 and Table R2 was added as the new Supplementary*
 *Figure S17 and Table S8, respectively.*

Figure R5. The stable configurations of (a-e) O₂, (f) CO and (g) Cl₂ adsorbed at the
 different sites of Fe/MnO₂ surface, and O₂ adsorbed in (h) the oxygen vacancy or (i)
 bi-manganese sites of MnO₂. The corresponding adsorption energy with zero-point
 energy correction is listed in the table of the picture.

Oxygen atom	a1	b1
Bader Charge	-0.12	-0.12

Figure R19. The difference charge density of O₂ adsorbed on Fe/MnO₂. Different
 oxygen atoms from adsorbed oxygen species were labeled as a1 and b1, and
 corresponding Bader charge was recorded in the bottom of charge density map. The
 charge density of yellow and blue represents the concentrated and scarce electrostatic
 potential scale respectively.

Table R2. Calculated energy and zero-point energy of the corresponding free and
 adsorbed molecules.

Configuration	Energy (eV)	Zero Point Energy (eV)	Corrected Energy (eV)
Fe-Fe-O ₂	-545.52	0.08	-545.44
Fe-O-O-Fe	-544.97	0.06	-544.91
Fe-Mn-O ₂	-545.25	0.07	-545.18
Fe-Mn-VO ₂	-547.20	0.05	-547.15
Mn-O ₂	-545.19	0.09	-545.10
Mn-CO	-551.88	0.12	-551.76
Mn-VO ₂ -1	-548.80	0.05	-548.75
Mn-VO ₂ -2	-548.13	0.08	-548.05
Cl ₂ -Mn	-540.66	0.01	-540.65
O ₂	-8.84	0.10	-8.74
CO	-14.80	0.13	-14.67
Cl ₂	-3.58	0.03	-3.55

**References cited in the Response Letter.**

- 1 Montemore, M. M., van Spronsen, M. A., Madix, R. J. & Friend, C. M. O₂
Activation by Metal Surfaces: Implications for Bonding and Reactivity on
Heterogeneous Catalysts. *Chem. Rev.* **118**, 2816-2862 (2018).
- 2 Mukherjee, D., Ellern, A. & Sadow, A. D. Remarkably Robust Monomeric
Alkylperoxyzinc Compounds from Tris(oxazolanyl)boratozinc Alkyls and O₂. *J.*
*Am. Chem. Soc.* **134**, 13018-13026 (2012).
- 3 Ma, Y. et al. Investigation into the Enhanced Catalytic Oxidation of o-Xylene
over MOF-Derived Co₃O₄ with Different Shapes: The Role of Surface
Twofold-Coordinate Lattice Oxygen (O_{2f}). *ACS Catal.* **11**, 6614-6625 (2021).
- 4 Li, H. et al. Analysis of the limitations in the oxygen reduction activity of
transition metal oxide surfaces. *Nat. Catal.* **4**, 463-468 (2021).
- 5. Nie, L. et al. Activation of surface lattice oxygen in single-atom Pt/CeO₂ for
low-temperature CO oxidation. *Science* **358**, 1419-1423 (2017).
- 6. Davó-Quiñonero, A. et al. Insights into the Oxygen Vacancy Filling Mechanism
in CuO/CeO₂ Catalysts: A Key Step Toward High Selectivity in Preferential CO
Oxidation. *ACS Catal.* **10**, 6532-6545 (2020).
- 7 Liu, J.-X., Su, Y., Filot, I. A. W. & Hensen, E. J. M. A Linear Scaling Relation
for CO Oxidation on CeO₂-Supported Pd. *J. Am. Chem. Soc.* **140**, 13624 (2018).
- 8 Yang, J. et al. Oxygen Vacancy Promoted O₂ Activation over Perovskite Oxide
for Low-Temperature CO Oxidation. *ACS Catal.* **9**, 9751-9763 (2019).
- 9 Puigdollers, A. R. & Pacchioni, G. CO Oxidation on Au Nanoparticles Supported
on ZrO₂: Role of Metal/Oxide Interface and Oxide Reducibility. *ChemCatChem*
**9**, 1119-1127 (2017).
- 10. Bae, W. K. et al. Highly Effective Surface Passivation of PbSe Quantum Dots
through Reaction with Molecular Chlorine. *J. Am. Chem. Soc.* **134**, 20160-20168
(2012).
- 11. He, C. et al. Recent Advances in the Catalytic Oxidation of Volatile Organic
Compounds: A Review Based on Pollutant Sorts and Sources. *Chem. Rev.* **119**,
4471-4568 (2019).

- 12. Dai, Q. et al. HCl-Tolerant H_xPO_4/RuO_x-CeO_2 Catalysts for Extremely Efficient
Catalytic Elimination of Chlorinated VOCs. *Environ. Sci. Technol.* **55**,
4007-4016 (2021).
- 13 Nelson, N. C., Nguyen, M.-T., Glezakou, V.-A., Rousseau, R. & Szanyi, J.
Carboxyl intermediate formation via an in situ-generated metastable active site
during water-gas shift catalysis. *Nat. Catal.* **2**, 916-924 (2019).

REVIEWERS' COMMENTS

Reviewer #1 (Remarks to the Author):

The authors have provided more experimental evidence and calculation results to address my previous concerns, and have revised the manuscript and its SI accordingly. The whole manuscript has been improved drastically. Therefore, I would endorse its acceptance.

Reviewer #2 (Remarks to the Author):

The authors made very good effort for revising manuscript. The results structural characterizations provided in the revised manuscript and supporting information are satisfactory and I'm convinced by the authors' explanation. My one last suggestion is to include the information regarding specific activity at T50 in addition to the activation energies reported already in the manuscript since the mass of Fe is now clear through this revision thanks to the authors' effort for identifying the amount of Fe atoms on MnO₂, and discuss it with reported values (see for example, <https://www.nature.com/articles/s41929-019-0282-y> which summarizes the specific activities of single and few atom catalysts). I consider that the specific activity is one of the important figure of merit of catalyst to consider the practical applications.

Reviewer #3 (Remarks to the Author):

I will defer to the other reviewers on their comments, though they have made some very good points regarding the experimental data and its interpretation. I very much appreciate the authors' work to address the Reviewer 3 comments but I still have some reservations regarding the work as presented though these are fairly minor and should be easily addressed. First, the new term "adjacent single-atom iron" is still somewhat problematic. MnO₂-hosted Fe dimer seems like a much more appropriate term given that in the pathways presented, both iron atoms play some direct role, making the "single-atom" phrase inappropriate. I don't understand the continued inclusion in this manuscript. It also looks from the figures that the Fe is integrated into the MnO₂ oxide which means these are likely substitutional defects and not adatom like structures. This needs to be made more explicitly clear since "adjacent Fe atoms anchored on MnO₂" sounds like dimer pairs of adatoms which does not appear to be the structures considered since oxygen atoms are above the Fe (e.g., Figs. R19 and R5).

Also, more detail is required on the zero-point energy correction (e.g., in what limit was it calculated/ methodology used and which atoms were utilized for calculating... likely only the adsorbing molecule based on values but this should be explicitly stated). The treatment of end-on vs. side-on is appropriate, showing relevant DFT values and the additional DRIFTS argument regarding dipole makes some sense, though having a complimentary end-on O₂ plot analogous to R19 would be beneficial for showing there is a dipole shift in such cases, strengthening the dipole argument.

Response Letter

We thank all three Reviewers for their positive reviews and the constructive suggestions that help to improve the scientific presentation of this manuscript. Point-by-Point responses to address the concerns raised by the Reviewers are shown below:

Reviewer #1 (Remarks to the Author):

The authors have provided more experimental evidence and calculation results to address my previous concerns, and have revised the manuscript and its SI accordingly. The whole manuscript has been improved drastically. Therefore, I would endorse its acceptance.

Response: We sincerely thank the Reviewer for the approval of this manuscript.

Reviewer #2 (Remarks to the Author):

The authors made very good effort for revising manuscript. The results structural characterizations provided in the revised manuscript and supporting information are satisfactory and I'm convinced by the authors' explanation. My one last suggestion is to include the information regarding specific activity at T50 in addition to the activation energies reported already in the manuscript since the mass of Fe is now clear through this revision thanks to the authors' effort for identifying the amount of Fe atoms on MnO₂, and discuss it with reported values (see for example, <https://www.nature.com/articles/s41929-019-0282-y>, which summarizes the specific activities of single and few atom catalysts). I consider that the specific activity is one of the important figure of merit of catalyst to consider the practical applications.

Response: Thank you for your nice comment and helpful suggestion. Referring to the remarkable article [*Nat. Catal.*, 2019, 2, 590–602], we calculated the specific activity (R_{T50}) of 0.25% Fe/MnO₂ according to the calculation formula (1) and the value was 310 mol_{CO} h⁻¹ mol_{Fe}⁻¹ with T50 at 47 °C, higher than those of most platinum group

metal (PGM) catalysts.

Specific activity was calculated by the following equation:

$$R_{T50} = \frac{v_{CO} \times M_{Fe}}{m_{Fe}} \quad (1)$$

Where v_{CO} represents the molar gas flow rate of CO (mol/h); M_{Fe} is the relative atomic mass of Fe; m_{Fe} is the mass of Fe on the basis of ICP results and statistical results from 0.25% Fe/MnO₂ STEM image.

During the revision, we also cited this relevant literature [Nat. Catal., 2019, 2, 590–602] as Ref. 18 and added the corresponding discussion on page 20 in the revised manuscript as follows. “Simultaneously, we calculated the specific activity (R_{T50}) of 0.25% Fe/MnO₂ according to the calculation formula (10) and the value was 310 mol_{CO} h⁻¹ mol_{Fe}⁻¹ with T50 at 47 °C, higher than those of most platinum group metal (PGM) catalysts.^{[18]”}

Reviewer #3 (Remarks to the Author):

I will defer to the other reviewers on their comments, though they have made some very good points regarding the experimental data and its interpretation. I very much appreciate the authors’ work to address the Reviewer 3 comments but I still have some reservations regarding the work as presented though these are fairly minor and should be easily addressed. First, the new term “adjacent single-atom iron” is still somewhat problematic. MnO₂-hosted Fe dimer seems like a much more appropriate term given that in the pathways presented, both iron atoms play some direct role, making the “single-atom” phrase inappropriate. I don’t understand the continued inclusion in this manuscript. It also looks from the figures that the Fe is integrated into the MnO₂ oxide which means these are likely substitutional defects and not adatom like structures. This needs to be made more explicitly clear since “adjacent Fe atoms anchored on MnO₂” sounds like dimer pairs of adatoms which does not

appear to be the structures considered since oxygen atoms are above the Fe (e.g., Figs. R19 and R5).

Also, more detail is required on the zero-point energy correction (e.g., in what limit was it calculated/ methodology used and which atoms were utilized for calculating... likely only the adsorbing molecule based on values but this should be explicitly stated). The treatment of end-on vs. side-on is appropriate, showing relevant DFT values and the additional DRIFTS argument regarding dipole makes some sense, though having a complimentary end-on O₂ plot analogous to R19 would be beneficial for showing there is a dipole shift in such cases, strengthening the dipole argument.

Response: We thank the reviewer for the constructive advices and would like to reply to the comments as follows.

(1) In order to avoid the confusion, we added *“also called as MnO₂-hosted Fe dimer,”* in the abstract to further explain the structure of catalyst developed in this study.

(2) Zero-point energy correction was obtained from vibrational frequencies by applying normal-mode analysis through density functional theory calculations [*J. Am. Chem. Soc.* 2021, 143, 1399-1408]. We fixed the catalyst substrate and only allowed the adsorbing molecule to vibrate.

(3) We added the configuration of oxygen molecule adsorbed on the two adjacent Fe sites with end-on mode, and Bader charge calculation results revealed the two oxygen atoms had different charges (-0.37 and -0.06) (Figure R1a). The different charges constantly changed the dipole moment of O₂ in the vibration process, accounting for the infrared characteristic absorption peak of Fe(O=O)Fe. However, for the side-on configuration (Figure R1b), two oxygen atoms got the same number of electrons, and the electron density was evenly distributed among the adsorbed oxygen molecule. Therefore, the dipole moment of oxygen molecules did not change during the vibration process. As a result, the adsorption configuration of oxygen on Fe sites was determined with end-on mode.

Figure R1. The difference charge density of O₂ adsorbed on Fe/MnO₂ with (a) end-on and (b) side-on mode. Different oxygen atoms from adsorbed oxygen species were labeled as a and b, and corresponding Bader charge was recorded in the bottom of charge density map. The charge density of yellow and blue represents the concentrated and scarce electrostatic potential scale respectively.

During the revision, Figure R1 was added as the Figure 3d and the Supplementary Figure S18. The discussion about zero-point energy correction was added on page 6 in the revised manuscript as follows. “Zero-point energy correction was obtained from vibrational frequencies by applying normal-mode analysis through density functional theory calculations.^[43] We fixed the catalyst substrate and only allowed the adsorbing molecule to vibrate.”

References cited in the Response Letter.

- [1] Beniya, A. & Higashi, S. Towards dense single-atom catalysts for future automotive applications. *Nat. Catal.* **2**, 590-602 (2019).
- [2] Kim, J. *et al.* Tailoring binding abilities by incorporating oxophilic transition metals on 3D nanostructured Ni arrays for accelerated alkaline hydrogen evolution reaction. *J. Am. Chem. Soc.* **143**, 1399-1408 (2021).